# Epitranscriptomic subtyping, visualization, and denoising by global motif visualization

Jianheng Liu [1,2,4] ✉, Tao Huang[3,4], Jing Yao[1,4], Tianxuan Zhao[1], Yusen Zhang[1] & Rui Zhang [1] ✉

Advances in sequencing technologies have empowered epitranscriptomic profiling at the single-base resolution. Putative RNA modification sites identified from a single high-throughput experiment may contain one type of modification deposited by different writers or different types of modifications, along with false positive results because of the challenge of distinguishing signals from noise. However, current tools are insufficient for subtyping, visualization, and denoising these signals. Here, we present iMVP, which is an interactive framework for epitranscriptomic analysis with a nonlinear dimension reduction technique and density-based partition. As exemplified by the analysis of mRNA m⁵C and ModTect variant data, we show that iMVP allows the identification of previously unknown RNA modification motifs and writers and the discovery of false positives that are undetectable by traditional methods. Using putative m⁶A/m⁶Am sites called from 8 profiling approaches, we illustrate that iMVP enables comprehensive comparison of different approaches and advances our understanding of the difference and pattern of true positives and artifacts in these methods. Finally, we demonstrate the ability of iMVP to analyze an extremely large human A-to-I editing dataset that was previously unmanageable. Our work provides a general framework for the visualization and interpretation of epitranscriptomic data.

The recent discovery of various RNA modifications in the transcriptome has given rise to the fast-growing field of epitranscriptomics. Individual RNA modification sites have been found to regulate nearly all aspects of RNA metabolism and are involved in a wide range of biological processes[1,2]. Accordingly, RNA modification writers are spatiotemporally controlled, and their misregulation leads to a number of pathologies. For example, the writers of m⁶A and A-to-I editing are misregulated in multiple types of cancers and have been identified as promising therapeutic targets of cancers[3–5], with related cancer treatments in pre-clinic or pre-investigational new drug (IND) stages. Catalytic inhibitor of METTL3 STM2457 leads to impaired engraftment and prolonged survival in various mouse models of AML[5].

ADAR1p150 inhibitor Rebecsinib prevents malignant A-to-I editing-mediated leukemia stem cell self-renewal in completed pre-IND studies[6].

Methods for transcriptome-wide RNA modification detection at the single-base resolution are undergoing rapid development. The common strategies are to convert RNA modification signals into next-generation sequencing-detectable signals, such as mutation and truncation (e.g.[7–11]). Putative RNA modification sites called by these strategies from a single high-throughput experiment contain one type of modification deposited by different writers (e.g., m⁵C called from RNA BS-seq approach or pseudouridine called from CMC-related methods[12–14]) or different types of modifications (e.g., A-to-I editing

[1]MOE Key Laboratory of Gene Function and Regulation, Guangdong Province Key Laboratory of Pharmaceutical Functional Genes, State Key Laboratory of Biocontrol, School of Life Sciences, Sun Yat-Sen University, Guangzhou 510275, P. R. China. [2]Department of Pharmacology, Weill Cornell Medicine, Cornell University, New York, NY 10065, USA. [3]Department of Pathology and Pathophysiology, Shantou University Medical College, Shantou 515041, P. R. China. [4]These authors contributed equally: Jianheng Liu, Tao Huang, Jing Yao. ✉e-mail: liujh26@mail2.sysu.edu.cn; zhangrui3@mail.sysu.edu.cn

and m[1]A called from RNA-seq[11,15] or m[6]A and m[6]Am called from m[6]A antibody-dependent methods[16]). Meanwhile, various levels of noise are present in such putative RNA modification sites because most methods have limited accuracy and specificity[17]. Despite the importance of the precise classification of modification types and their corresponding writers, the currently developed tools are insufficient to address this issue.

Most types of RNA modifications occur within a specific sequence and/or structural context[18–23]. Thus, we may assign the writers of each authentic modification site even without experimental validation, as long as prior knowledge of writer target motifs is provided. Typically, we process the putative modification sites and flanking sequences by using motif finders[24], which utilize probability statistics, graph theory, deep learning, or other approaches, to obtain a set of enriched motifs, and then determine the writers of some modification sites based on their motifs. However, such a process is unable to estimate the levels of noise within the dataset. Moreover, current motif finders are slow or fail to manage inputs with over 10,000 records, such as RNA modification sites called from RNA-seq data or Nanopore direct RNA-seq data[25,26]. We realize that the goal of analyzing RNA modifications and noise based on sequence features is, in some ways, similar to the high-resolution dissection of tissue compositions using single-cell RNA-seq technologies. Hence, we may take inspiration from the strategies used to analyze the large numbers of parameters generated in single-cell studies for RNA modification analysis.

Here we developed a framework termed interactive epitranscriptomic Motif Visualization and Subtype Partition (iMVP) for epitranscriptomic subtyping, visualization, and denoising, with demonstrations of its utility on various kinds of high-throughput experiments.

## Results

### Establishment of iMVP

Our strategy is based on two principles. (1) Most RNA modifications are deposited within a specific sequence context, which is determined by the biophysical interactions between writers and their substrates. This is presumably also true for most artifacts introduced by the enzymic and chemical processes in modification detection. (2) The input sequences only account for a tiny subset of the k-mer space. Accordingly, our iMVP framework consists of five steps for the analysis and visualization of the topological distribution of RNA modifications, as follows (Fig. 1a): (1) the k-mer patterns of the modification sites were extracted and encoded into a computer-readable format-here, we one-hot encoded the 21-mer sequences surrounding the modification sites; (2) a dimension reduction algorithm was used to project the extracted patterns onto a 2-D plane, where the similarities of the sequences were therefore approximated by the Euclidean distances in the projections; (3) an unsupervised clustering algorithm was applied to group the enriched patterns in the projections; (4) clusters were extracted automatically or manually based on the contours via our interactive interface with the help of visualization; and (5) further analyses of the clusters, including (but not limited to) drawing the logos with WebLogo[27] or discovering motifs by canonical methods (e.g. MEME[28]), were performed.

We started with dimension reduction tools that are widely used in single-cell analyses because their properties and performances have been well evaluated[29,30], especially t-distributed Stochastic Neighborhood Embedding (t-SNE)[31] and Uniform Manifold Approximation and Projection (UMAP)[30]. To benchmark the candidate algorithms, we selected m[5]C profiling data in fly embryos, which consisted of experimentally validated NSUN2-dependent (Type I, 90.9%) sites and NSUN6-dependent (Type II, 9.1%) sites with clear motif preferences (Supplementary Data 1)[21]. As expected, in all tested algorithms, RNA modification sites with a stringent sequence requirement, i.e., Type II

sites with a strong 5′-CUCCA-3′ motif[21], were highly condensed (Fig. 1b); in contrast, sites with a less stringent sequence requirement, i.e., Type I sites[19,32], were projected into a broad area with a continuous density distribution (Fig. 1b). To quantitatively evaluate the candidate algorithms, we developed two metrics, the outgroup-ingroup score and boundary score (see "Methods" section), to measure their condensability and discernibility, respectively. We found that t-SNE, Open t-SNE[33], and UMAP had better condensabilities (i.e., higher outgroup-ingroup scores) than PCA, Isomap[34], and DensMAP[35], generating projections much closer to their corresponding centroids. Compared with t-SNE and Open t-SNE, UMAP had a better discernibility (i.e., a lower boundary score), making the boundaries of Type I and Type II sites much clearer (Fig. 1b and Supplementary Fig. 1a). These findings were also true for more complicated m[5]C data from nocodazole (Noc) treated HeLa cells, in which at least 4 motifs, including two minor motifs, are present (Supplementary Fig. 1b). We also tested UMAP with different sequence encoding methods, such as PCA preprocessing (a regular step in single-cell analysis[36]) and numeric labeling. No other methods with better performance than that of the one-hot encoding method were found (Supplementary Fig. 1c). Additionally, we tested UMAP and t-SNE on different initialization methods with a simulation dataset containing 109,850 sequences (Supplementary Note 1). We found that most motifs can be correctly clustered by UMAP and t-SNE, no matter which method is used (Supplementary Fig. 1d). Last, we tested UMAP with different random seeds or metric functions (Supplementary Fig. 1e, f), and it demonstrated consistent robustness across different parameter settings in general. Since UMAP and t-SNE exhibited comparable performance, we opted for UMAP for the subsequent analyses.

We next sought a suitable unsupervised clustering algorithm based on the UMAP output. Among the tested algorithms, we found that both density-based algorithms (e.g., HDBSCAN[37], DBSCAN[38], and OPTICS[39]) and graph-based algorithms (e.g., Spectral clustering[40], Louvain[41], and Leiden[42]) were able to recognize distinct clusters from UMAP projections (Fig. 1d and Supplementary Fig. 2a). Among these methods, HDBSCAN, Leiden, and Louvain demonstrated acceptable computational cost (Fig. 1e) and exhibited superior performance in clustering, as indicated by precision, recall, and Adjusted Rand Index (ARI) metrics (Fig. 1f, Supplementary Fig. 2b, and Supplementary Note 1). While Louvain and Leiden achieved higher ARI scores in fly dataset (Supplementary Fig. 2b), HDBSCAN outperformed them when handling noisy datasets that contain randomly generated sequences (Supplementary Fig. 2c-e). Moreover. Louvain and Leiden were found to be less flexible with only one adjustable parameter and tended to generate interleaved small clusters (Supplementary Note 2 and Supplementary Fig. 2f). In contrast, HDBSCAN enables both standard clustering and "soft clustering" (complete assignment), which makes it more flexible in different scenarios.

Since the majority of reported RNA modification motifs are within a range of 15 nucleotides (nt)[1], we chose to analyze 21-nt sequences in our computations. Furthermore, we conducted tests using k-mers of varying lengths and observed that iMVP demonstrated robustness in handling k-mers ranging from 11 to 51 nt (Supplementary Fig. 3). In combination with UMAP and HDBSCAN, iMVP can process >8,000 21-mer sequences within 3 min. For ease of use, we also developed an interactive interface for iMVP (Supplementary Movie 1), which allows us to combine the automatic partitioning results with manually selected clusters. Both the cookbooks for iMVP and the interactive interface are available on GitHub (https://github.com/SYSU-zhanglab/iMVP).

### Comparison of iMVP with traditional tools

To compare the performance of iMVP with other traditional tools in motif search, we analyzed two simulation datasets that comprised 100,000 and 200 sequences with 12 and 5 heterogeneous motifs,

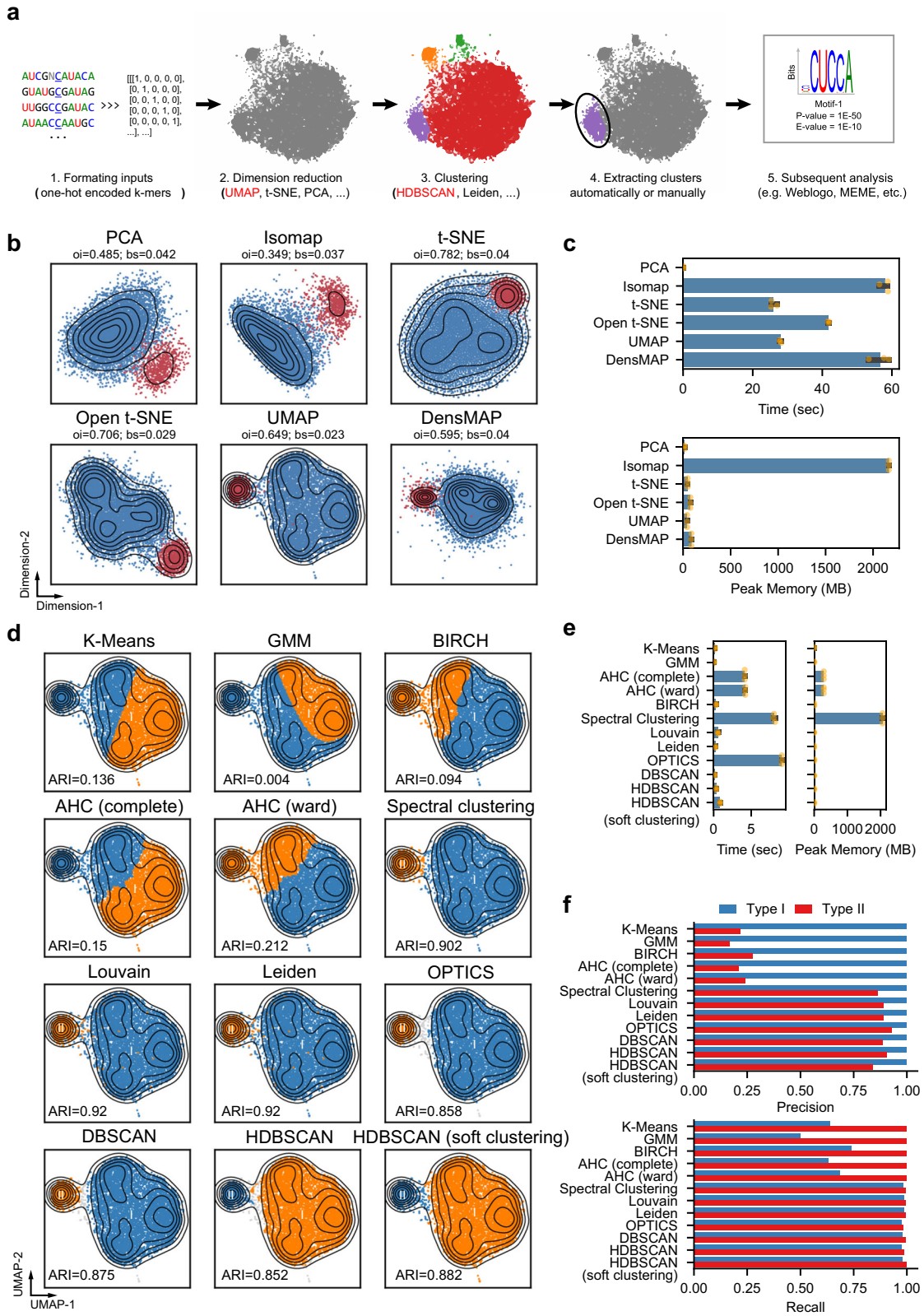

respectively (Supplementary Note 1). Our initial comparison was between iMVP and MEME. In the larger dataset, iMVP demonstrated superior ability to identify less abundant motifs compared to MEME, whereas both methods performed similarly in the smaller dataset (Supplementary Fig. 4–6 and Supplementary Note 3).

Next, to ensure a fair comparison between iMVP and traditional tools (including MEME, STREME, and HOMER), we extensively optimized the parameters of the traditional tools for the larger dataset, resulting in the generation of 200 parameter sets (see "Methods" section). We evaluated the performance of different methods by analyzing their computational time and accuracy in motif searching. Overall, our findings revealed that iMVP surpassed existing tools in terms of both computational time and precision (Supplementary Fig. 7).

**Fig. 1 | The establishment of iMVP. a** The schema for iMVP. The algorithms selected in our analysis are labeled in red. **b** Testing dimension reduction algorithms with fly embryo m5C dataset. Type I and Type II sites are labeled in blue and red, respectively. PCA, Principal Component Analysis; LLE, Local Linear Embedding; t-SNE, t-distributed Stochastic Neighbor Embedding; UMAP, Uniform Manifold Approximation and Projection; DensMAP, Density-Preserving UMAP. Random states were set to 42. Parallel levels were set to 4 if possible. Algorithm specific parameters were listed in Supplementary Data 7. Metrics: oi outgroup-ingroup score, higher is better, bs boundary score, lower is better. **c** The peak memory usage and runtime of the algorithms used in **b**. The bar plot displays the mean and 95% confidence interval, while the precise values of the data points are visualized as dots. n = 3 biologically independent experiments. The time consumed by JIT (Just-In-Time) compiling process in UMAP was not included. This process typically takes

-200 seconds. **d** Testing clustering algorithms based on the results of UMAP. Simple partitioning algorithms: K-Means and Gaussian Mixture Model (GMM); Hierarchical clustering: Agglomerative hierarchical clustering and BRICH; Graph-based: Spectral clustering, Louvain, and Leiden; Density-based: OPTICS, DBSCAN, and HDBSCAN. Random states were set to 42. Parallel levels were set to 4 if possible. Algorithm specific parameters were listed in Supplementary Data 7. **e** The peak memory usage and runtime of the algorithms used in **d**. The bar plot displays the mean and 95% confidence interval, while the precise values of the data points are visualized as dots. n = 3 biologically independent experiments. For Louvain and Leiden, the time (but not memory usage) of the construction of Nearest Neighbor Matrix with UMAP was also included. **f** The Precision and Recall rate of the clustering algorithms in **d**. Precision and Recall were computed based on our experimentally validated site information of the dataset.

## Relaxation of position-dependency of iMVP to broaden its applications

In addition to the scenario in which positions with modification-induced signals matched the coordinates of RNA modification sites themselves, signals generated due to the modified bases can be shifted to adjacent positions. For example, such an issue is common in signals from Nanopore direct RNA-seq, which shows promise for discriminating and identifying different RNA modifications in native RNA. Several methods based on base-calling errors, altered current intensities, or trace profiles of Nanopore direct RNA-seq data have been developed to identify signatures that represent RNA modifications (e.g[43–46].). With these methods, modification caused errors and disturbances have been found to occur from the −3 to +3 positions relative to the modification sites[43,46]. Because of the position-aware nature of iMVP, we proposed a phase-matching strategy to solve such a "phase mismatching" issue, thus broadening the applications of iMVP.

To demonstrate this strategy, we analyzed the RNA modification sites called by xPore based on the differential Gaussian distribution of Nanopore signal parameters between wild type and METTL3 knockout HEK293T cells[26] (Fig. 2a). In this dataset, RNA modification signals tended to locate in the −1 to +1 positions adjacent to the modified bases (three phases)[26] (Supplementary Fig. 8a). To perform phase matching, we first applied iMVP to the unprocessed xPore variant data and grouped the sites into 9 clusters (Supplementary Data 2). Clusters #1 to #3, which accounted for 47.9% of the sites (3,446 sites), were within RRACH (R = A/G, H = A/C/U) motifs in different phases (Fig. 2b). To match the phases of m6A motifs, we recentered the sequences of cluster #1 to cluster #3 onto an "A" if possible (Fig. 2c). Other small clusters were centered by non-A bases (Supplementary Fig. 8b–e) and were less likely to be m6A signals; thus, they were not included in the analysis. After phase matching, we defined 95% (3,175) of sites as authentic m6A sites within RRACH motifs (Fig. 2d). The remaining 5% of sites were within the CAR motif (Fig. 2d). Despite the CAR motif resembles previously reported m6Am motif (BCA (B = C/G/U)[47,48] or BBCABW[49], as METTL3 is exclusively associated with m6A modification, these sites are likely to be false positives. Consistently, when examining the locations of CAR sites relative to transcription start site (TSS), no enrichment around TSS was observed (Supplementary Fig. 8f).

To evaluate the reliability of putative modification sites defined using iMVP, we applied a different method, i.e., m6A-seq approach, to identify m6A/m6Am in the same cell line (see "Methods" section). A winscore-based method[50] was used to quantify the m6A/m6Am peaks in our m6A-seq dataset. Winscore quantifies the enrichment of modified reads within a 50-nt window relative to gene expression and modified reads across the entire gene. As a positive control, we used m6A/m6Am sites identified by m6ACE-seq (Supplementary Fig. 9)[51]. We found that m6A clusters in both xPore and m6ACE-seq datasets had significantly higher winscores than the background winscores, indicative of authentic m6A sites (Fig. 2e). However, m6Am-like clusters in m6ACE-

seq dataset but not xPore dataset had significantly higher winscores (Fig. 2e). This result is consistent with the role of METTL3 in regulating only m6A sites and suggests that the previously reported xPore sites with CAR motifs were likely false positives. Our analysis highlights a phase-matching strategy to expand the applications of iMVP.

To further relax the position-dependency of iMVP to broaden its applications, we introduced a sliding window strategy. This strategy consists of three steps: (1) generating sliding windows on input sequences; (2) performing iMVP on all the windows to extract clusters; and (3) conducting motif alignment and comparison on the extracted clusters to identify the patterns. To test this strategy, we utilized a simulation dataset comprising 500 PDX1 motifs randomly distributed in a 50-bp sequence, along with 500 random noise sequences. We first generated 20-bp windows with a 1-bp step, resulting in a total of 30,000 windows. Then we applied iMVP to these 30,000 windows (Supplementary Fig. 10a, b) and extracted enriched patterns (clusters #1 to #11) (Supplementary Fig. 10c). Among these patterns, 10 exhibited distinct PDX1 motifs. These motifs may be further aligned with Tomtom to generate a consensus motif (Supplementary Fig. 10d). By following all three steps, we identified between 48 to 483 sequences containing PDX1 motifs. The false discovery rate (FDR) ranged from 7.7% to 45.5%, depending on the specific thresholds applied (Supplementary Fig. 10e–g).

## iMVP assists in the discovery of previously unknown mRNA m5C writers and known writer divergence across species

For the well-studied RNA modifications, such as m6A, m5C and pseudouridine, it is known that multiple writers have evolved to target unique subsets of the modification sites based on the differential protein-RNA biophysical interactions. The map of a specific RNA modification type is built upon this principle, and in turn, we can infer the writers of this map by visualizing the sequence features of all modification sites of this map. Foreseeing the needs in novel writer mining, we employed iMVP to investigate the most comprehensive m5C profiles from various samples in humans, mice, frogs, zebrafish, and flies[19,32], where new writers may be hidden. Intriguingly, in addition to two clusters that represent known Type I and Type II sites, two more clusters appeared in early developmental stage samples (Fig. 3a, Supplementary Fig. 11, and Supplementary Data 3) and 48-hour Noc-treated HeLa cells (Fig. 3b and Supplementary Data 4), which are known to be enriched with mRNA m5C[32]. The first cluster (termed Type III sites) was found in both early developmental stages in vertebrates (Fig. 3a and Supplementary Fig. 11) and Noc-treated HeLa cells (Fig. 3b). This cluster contained a 5′-GUNGCCANNUG-3′ motif in a less structured sequence and accounted for 0.67% to 2.8% of the sites. The second cluster (termed Type IV sites) was only found in Noc-treated HeLa cells (Fig. 3a, b and Supplementary Fig. 11). This cluster contained a 5′-UUCGANGU-3′ motif and accounted for 11.04% of the sites. Notably, these 4 types of m5C sites cannot always be detected by MEME and HOMER (Supplementary Fig. 12).

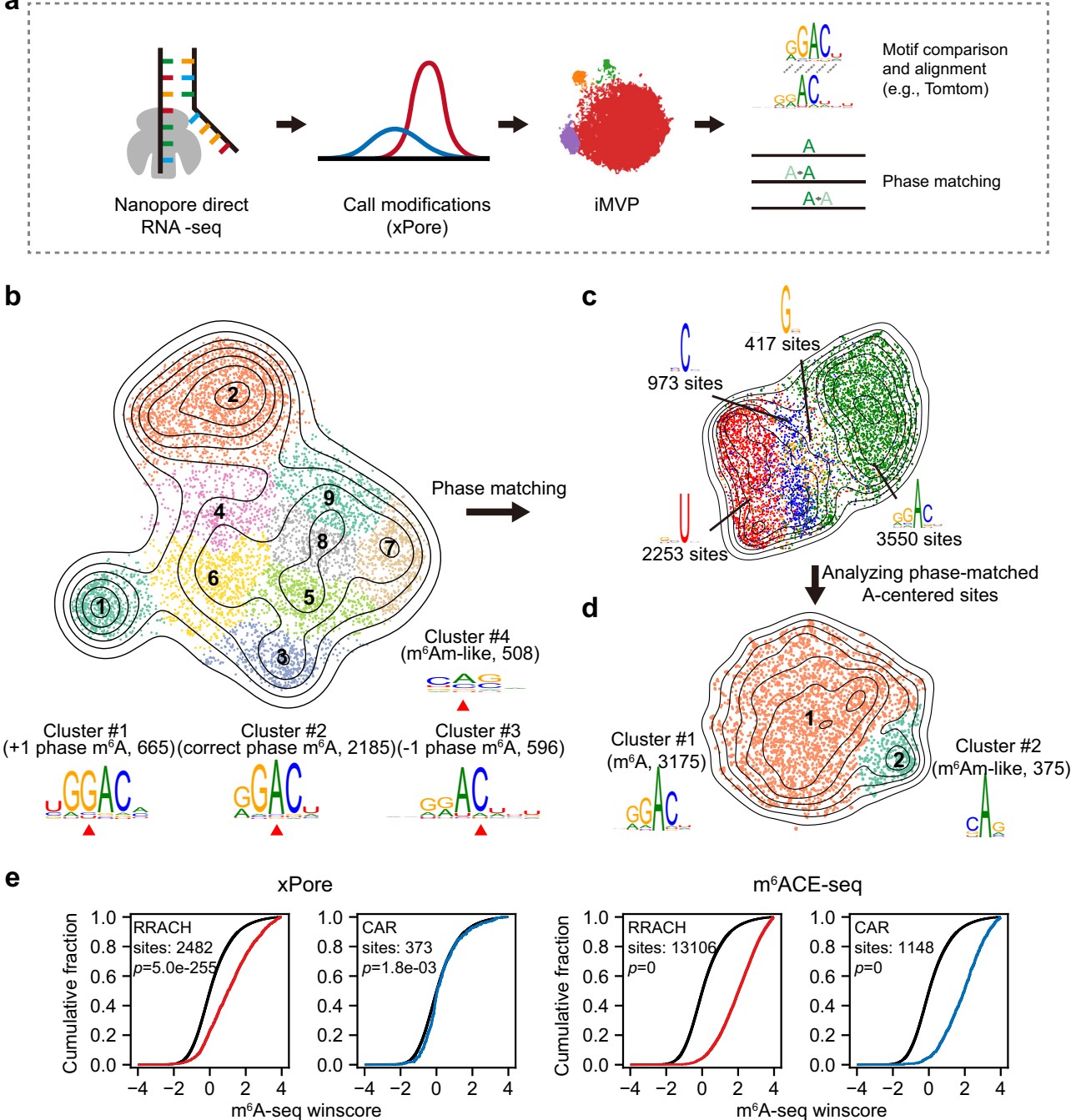

**Fig. 2 | A phase-matching strategy to broaden the applications of iMVP in Nanopore direct RNA-seq. a** The diagram showing the Nanopore-xPore-iMVP strategy with phase matching. **b** The global visualization of xPore output without phase matching. Clusters #1 to #3 are m6A-like motifs in different phases. The bases in the center of the k-mers were indicated by red arrowheads. **c** The global visualization of phase-matched xPore output. Four types of center bases were shown in different colors. **d** The global visualization of the phase-matched A-centered cluster in **c**. **e** Cumulative distributions of m6A-seq winscores of RRACH cluster and CAR cluster from m6ACE-seq and xPore datasets (see "Methods" section). In each subplot, the windows of RRACH and CAR clusters were highlighted in red and blue, respectively, and input windows were in black. Windows with reads per kilobase per million mapped reads (RPKM) values of ≥1 in the input were used for analysis. The *P* values were determined using a two-sided Kolmogorov–Smirnov test. One sample with one IP and one input experiment was performed to obtain winscore data.

The sequence contexts of Type III and IV sites resembled the C2278 (Fig. 3c) and C2870 (Fig. 3d) sites in yeast 25 S rRNA methylated by Rcm1 (NSUN5) and Nop2[52], suggesting that these two proteins may be new mRNA m5C writers. To validate this inference, we knocked out NSUN5 (Supplementary Fig. 13a) and knocked down Nop2 via siRNA in HeLa cells (Supplementary Fig. 13b). As a control, NSUN2 and NSUN6 were knocked out separately. As expected, loss or decreased levels of m5C methylation in each cluster was observed in the corresponding

knockout or knockdown cells (Fig. 3e and Supplementary Fig. 13c). Consistently, we found that sites with a Type III motif had increased m5C levels (Supplementary Fig. 13d-e and Supplementary Data 4) when re-analyzing previously published NSUN5 overexpression data from LN229 cells[53]. Interestingly, upon the overexpression of NSUN5, a minor Type III sub-motif, 5′-GUNNNCCAKHUG-3′ (K = A/C), was also found (Supplementary Fig. 13f). Taken together, these findings indicate that NSUN5 and Nop2 are previously unknown mRNA m5C writers,

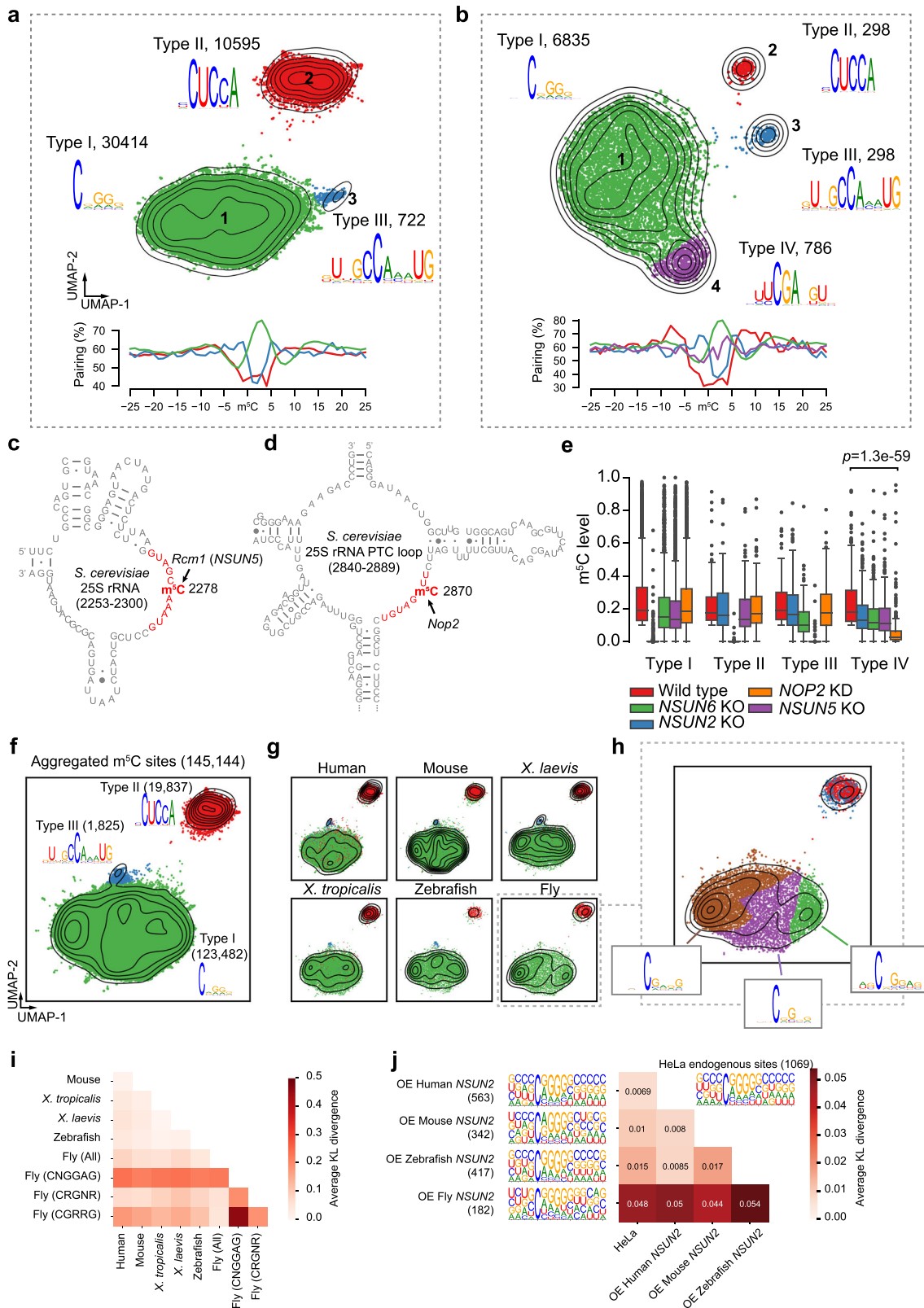

suggesting that our global motif analysis strategy enables new motif and writer discovery.

Global visualization may also facilitate an intuitive comparison of differences in motifs among different biological contexts. To determine whether iMVP can assist with such a comparison, we examined the motif divergence of different types of m5C sites across species with iMVP. We first aggregated m5C sites (145,144 sites in total) from six

species to generate a combined iMVP map. Next, we projected the sites of individual species onto the combined map for comparison. Interestingly, despite all three types of sites being highly consistent among all six species (Fig. 3f, g), a motif preference divergence between fly and vertebrate Type I sites was observed, wherein fly NSUN2 had a stronger preference for the CGRRG and CNGGAG motifs (Fig. 3g, h). Compared with the other two sub-types of motifs found in fly,

**Fig. 3 | The discovery of novel mRNA m⁵C writers and known writer divergence across species with iMVP. a, b** The global visualization of the m⁵C sites identified in human oocytes (**a**) and Noc-treated HeLa cells (**b**). Different clusters were shown in different colors. The motifs and the metaprofiles of the secondary structure for each cluster were shown. The colors of the metaprofiles followed those in the UMAP projections. **c, d** The C2278 and C2870 m⁵C sites on yeast 25 S rRNA methylated by Rcm1 (NSUN5) and Nop2, respectively. The inferred recognition motifs of NSUN5 and Nop2 were highlighted in red. The structure views were adopted from Cannone et al.[69]. **e** The methylation level changes of the four types of m⁵C sites in wild type and NSUN family member KO/KD HeLa cells treated with Nocodazole. $n = 1$ for each type of cells. Sites covered by at least 20 reads in all samples and with methylation level ≥0.1 in wild-type cells were analyzed. The *P*-value was calculated using a one-sided student's t-test. Boxplots: 25th to 75th percentiles (boxes), medians (horizonal lines), and 1.5 times of the interquartile range (whiskers). **f** The global visualization of 145,144 aggregated m⁵C sites from all the species analyzed. Three distinct clusters and corresponding m⁵C types were shown. **g** The m⁵C site distributions of each species in **f**. **h** Subtyping the two major clusters in flies. Type I sites in flies were divided into three sub-types: left, CGRRG (R = A/G); middle, CRGNR; right, CNGGAG. Among these motifs, fly NSUN2 preferred CGRRG and CNGGAG, while vertebrate NSUN2 had no preference. **i** Pairwise comparison of Type I motifs (+1 to +5 positions) in different species. Metrics: Average KL divergence of the motifs. Lower values indicate a higher degree of similarity between two motifs. **j** Pairwise comparison of Type I motifs found in HeLa cells and those methylated by exogenously expressed NSUN2 from human, mouse, zebrafish, and fly in NSUN2 knockout HeLa cells. $n = 1$ for each type of cells. OE, over-expression.

vertebrate NSUN2 motifs exhibited a higher degree of similarity to the CRGNR sub-motif, as indicated by lower KL divergences (Fig. 3i). To investigate whether this difference was due to the preference of NSUN2, we overexpressed NSUN2 from human, mouse, zebrafish, and fly in NSUN2 knockout HeLa cells. The motifs found in this assay closely aligned with the finding in iMVP (Fig. 3j). Taken together, iMVP assists in the identification of subtle motif preference differences between vertebrate and fly NSUN2 proteins.

## iMVP facilitates the identification of false positives in RNA BS-seq data

One of the major challenges in epitranscriptomic studies is how to obtain a reliable map of RNA modifications. Of the methods developed for transcriptome-wide RNA modification mapping, most have limited accuracy and specificity, thus leading to considerable levels of false positives. Many artifacts introduced by the enzymic and chemical processes in modification detection, as well as errors generated during high-throughput sequencing, preferentially occur within certain sequence contexts. Thus, global motif visualization of the inputs may facilitate the identification of false positives.

To determine whether this is true, we applied iMVP to the m⁵C profiling data in adult human and mouse tissues. mRNA m⁵C sites in mammalian adult tissues appear to be much less frequent[19] and thus tend to be enriched in false positives, making accurately identifying real sites more challenging. Intriguingly, of the m⁵C profiles in human and mouse tissues, we indeed identified 4 (#4 to #7) and 2 (#4 and #5) minor clusters that were likely false positive, respectively (Fig. 4a, b). Both human and mouse clusters #4 were embedded in a homopolymer run of As that is prone to sequencing errors. The remaining human clusters (#5 to #7) were in primate-specific repeat element Alus. The second mouse cluster (#5) was in a GC-rich repetitive region that is difficult to be converted with BS-treatment. An examination of the raw mapping data revealed that these sites were clustered (Fig. 4c, d). Based on our finding that clustered sites in bisulfite sequencing tend to be false positives from conversion failure[19], these sites were believed to be artifacts that escaped from our filters due to close-to-threshold statistical parameters. These observations highlight the ability of iMVP to pinpoint possible false positives that were unaddressed by traditional filtering steps. Notably, iMVP itself can only identified enriched clusters, and additional experiment and/or knowledge is required to determine whether these clusters are false positives or not.

## Spiked iMVP distinguishes RNA-seq variants caused by RNA modifications from noise

Recent studies found that RNA modifications could introduce mismatches during reverse transcription; thus, variants called from RNA-seq may represent DNA mutations, RNA editing sites, or RNA modifications[15,54]. Based on these findings, algorithms, such as ModTect[11], have been developed to analyze RNA-seq data to identify putative RNA modification sites. Due to their complexity, it is expected that a high-level of noise is present in putative RNA modification sites inferred from RNA-seq data. To assist in the analysis of such datasets, we developed a modified iMVP strategy, termed spiked iMVP, and applied it to the ModTect dataset.

In spiked iMVP, k-mers containing known modification signals (m¹A, m¹acp³Ψ, m³C, m⁴C, and m²₂G) were spiked into the putative RNA modification-induced variants to label the motif preferences of known RNA modifications (Fig. 5a and Supplementary Data 5). Next, variants with different reference bases were separately analyzed with iMVP and clusters were visualized (Fig. 5b–e). For example, in the A base group, we identified cluster #1, which contained 22 novel m¹A-like sites and 5 known m¹A spike-in sequences (Fig. 5b). In the U base group, we observed a GUG preference labeled by m¹acp³Ψ (cluster #2) (Fig. 5c). In the G base group, which contained the most abundant variants called by ModTect, variants were grouped into three major clusters (cluster #4, cluster #5, and cluster #6) (Fig. 5e). Of the three clusters, the m²₂G spiked cluster #5 was enriched with a 5'U signature (Fig. 5e). These high-confidence m¹A sites, and the novel sequence feature and narrowed-down list of m¹acp³Ψ and m²₂G identified with iMVP will facilitate future validation and functional studies. Note that the clusters containing known RNA modifications only accounted for a small fraction of the maps; and many of the remaining clusters were likely false positives (Fig. 5b, d and Supplementary Fig. 14): some were surrounded by Cs; some were within a tract of A/U dinucleotide repeats; and some were within a strong poly(A)-tract. Such variants were likely artifacts introduced by enzymatic and optical processes during library construction and sequencing. In summary, Spiked iMVP adds a new layer of filter and helps pinpoint more reliable RNA modification candidates from datasets with substantial fractions of false positives.

## iMVP advances our understanding of m⁶A/m⁶Am profiles

A number of biochemical methods were recently developed to map transcriptome-wide m⁶A/m⁶Am at single-base resolution[10,26,51,55–58]. A systematical evaluation and comparison of sites called from different methods may shed light on our understanding of the comprehensive methylomes and the pattern of true positives and artifacts in these methods. However, such a comparison is missing due to the lack of tools. To bridge this knowledge gap, we collected m⁶A/m⁶Am sites from 7 studies with 8 different m⁶A/m⁶Am profiling methods (Fig. 6a and Supplementary Data 2): CIMS[10], CITS[10], m⁶ACE-seq[51], m⁶A-label-seq[55], MAZTER-seq[56], m⁶A-REF-seq[57], xPore[26], and DART-seq[58]. Globally, we found that only a small set of m⁶A and m⁶Am sites were overlapped among methods (Supplementary Fig. 15a), although the same cell type was used in all studies. This observation suggests that each method may only captured a subset of the methylated sites. Furthermore, the potential variability in false positive rates among different methods may also contribute to the observed low overlap. Thus, approaches based on site overlaps are not able to compare different methods and iMVP may be qualified for this task.

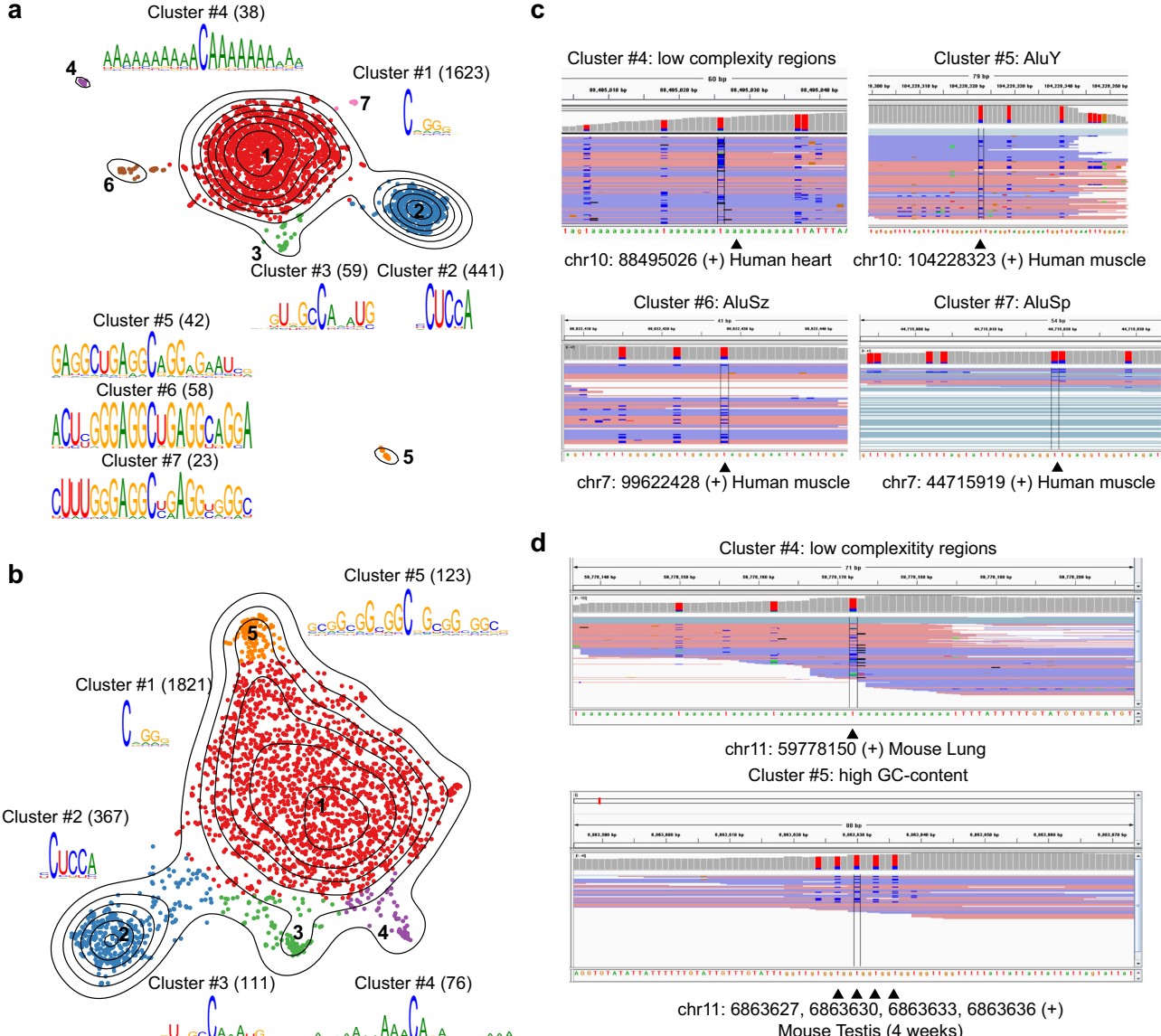

**Fig. 4 | The use of iMVP to identify noise in BS-seq data. a** Cluster #1 (Type I), cluster #2 (Type II) and cluster #3 (Type III) are canonical human m⁵C sites. Cluster #4 is artifacts from low-complexity regions. Clusters #5 to #7 are potential false positives from Alu repeats. 2,284 sites from human tissues were analyzed. **b** Cluster #1 (Type I), cluster #2 (Type II) and cluster #3 (Type III) are canonical mouse m⁵C sites, while cluster #4 and cluster #5 are potential false positives. 2,498 sites from mouse tissues were analyzed. **c** IGV browser view of selected human sites in clusters #4 to #7. Samples: human heart, GSM3462633; human muscle, GSM3462639. **d** IGV browser view of selected mouse sites in clusters #4 and #5. Samples: mouse lung, GSM3462647; mouse testis, GSM2461443.

We next performed iMVP on the aggregated 74,179 m⁶A/m⁶Am sites. Seven clusters were identified based on the patterns of density (Fig. 6b and Supplementary Fig. 15b). Clusters #1 to #4 were in canonical RRACH motifs, where cluster #1 had a strong 5′ A preference, and cluster #4 had no 5′ preference. Cluster #5 was a CAB motif, similar to previously reported m⁶Am motifs. Clusters #6 and #7 were in an A/G-rich context. In line with previous observations, the m⁶Am-like sites identified in Cluster #5 had the strongest enrichment around TSS; in contrast, the m⁶A sites in Cluster #1 to Cluster #4 had significantly weaker enrichments (Supplementary Fig. 15c). With global visualization, the extent of sequence preference in different methods was identified (Fig. 6c, d). Overall, CITS, m⁶ACE-seq, m⁶A-label-seq, and xPore had a broad range of k-mer types, while CIMS, MAZTER-seq, m⁶A-REF-seq, and DART-seq were biased towards certain types of k-mers. More specifically, CITS and m⁶ACE-seq, the two immunoprecipitation-based methods, had a similar k-mer distribution, despite CITS having a stronger enrichment in cluster #1(AA_A_CA). m⁶A-label-seq, which is

based on metabolic incorporation, had a unique enrichment in cluster #6. Although both MAZTER-seq and m⁶A-REF-seq utilized the MazF endoribonuclease strategy, sites identified by MAZTER-seq were more enriched in cluster #1. DART-seq, which is based on the hitchhiking of APOBEC C-to-U editing, had a strong bias in cluster #4 (NN_A_CU).

Since antibody-based methods represent an indirect way to infer m⁶A positions, antibody-independent approaches have become the recent focus of method development. Because the sensitivities of different methods vary and are dependent on the local sequence features of the modified bases, we evaluated the reliability of m⁶A-antibody-independent approaches by comparing the winscores of individual clusters with the background winscores using m⁶A-seq peak data (see "Methods" section). As a positive control, we used clusters identified by the m⁶A-antibody-dependent m⁶ACE-seq. DART-seq had the lowest m⁶A-peak enrichment among all methods (Fig. 6e), suggesting that many false positive signals may be introduced by APOBEC background editing. Compared with MAZTER-seq, signals from m⁶A-

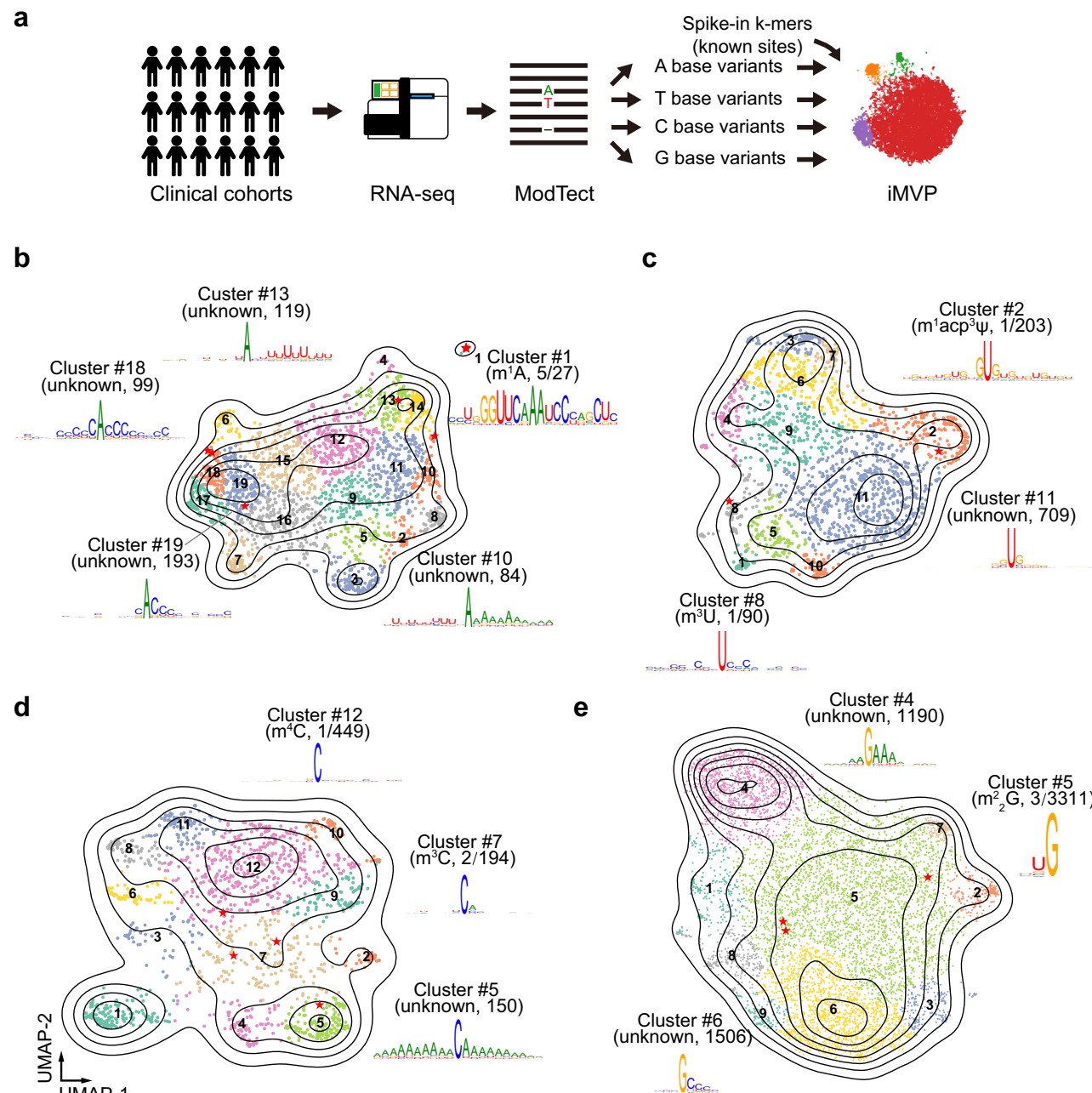

**Fig. 5 | Re-analysis of putative RNA modification induced variants identified by ModTect with Spiked iMVP. a** The diagram showing the Spiked iMVP strategy used in ModTect dataset analysis. **b**−**e** The global visualization of ModTect variants with A, U, C, and G as the reference base. The spiked sequences with known modifications, as well as the manually checked variants reported by Tan et al.[11], were shown as red asterisks. The motifs and the number of spike-in sites against total sites were also shown. 2522, 1912, 1564, and 7225 sites were analyzed separately.

REF-seq had lower m6A-peak enrichment (Fig. 6e), suggesting that the addition of FTO treatment might introduce noise. Overall, m6A-label-seq and MAZTER-seq had the highest m6A-peak enrichment for nearly all clusters, suggesting that they are currently the most reliable methods. These results provide valuable insights for the selection of m6A-antibody-independent approaches for m6A and m6Am profiles at single-base resolution.

## iMVP enables global motif discovery in an extremely large dataset

In canonical motif discovery tools, the input size is typically <10,000 sequences due to the high computational complexity of handling large datasets. Both UMAP and HDBSCAN are designed for large datasets,

and they are currently implemented in the NVIDIA RAPIDS library, which allows us to scale up and speed up iMVP analysis with GPU acceleration (Supplementary Fig. 16a, b). Thus, our framework may be capable of handling extremely large numbers of RNA modification sites that were previously unmanageable, such as millions of A-to-I RNA variants in the human genome.

To challenge iMVP, we analyzed the human A-to-I RNA editing list from REDIportal[25], which contains 15.6 million sites (5 million different 21-mers) (Fig. 7a). After dimension reduction, we reannotated the site list with UMAP coordinates to recover the densities. Globally, we observed distinct patterns of A-to-I RNA editing sites in Alu, repetitive non-Alu, and nonrepetitive regions (Supplementary Fig. 16c−e). Repetitive non-Alu and nonrepetitive sites were more centralized in

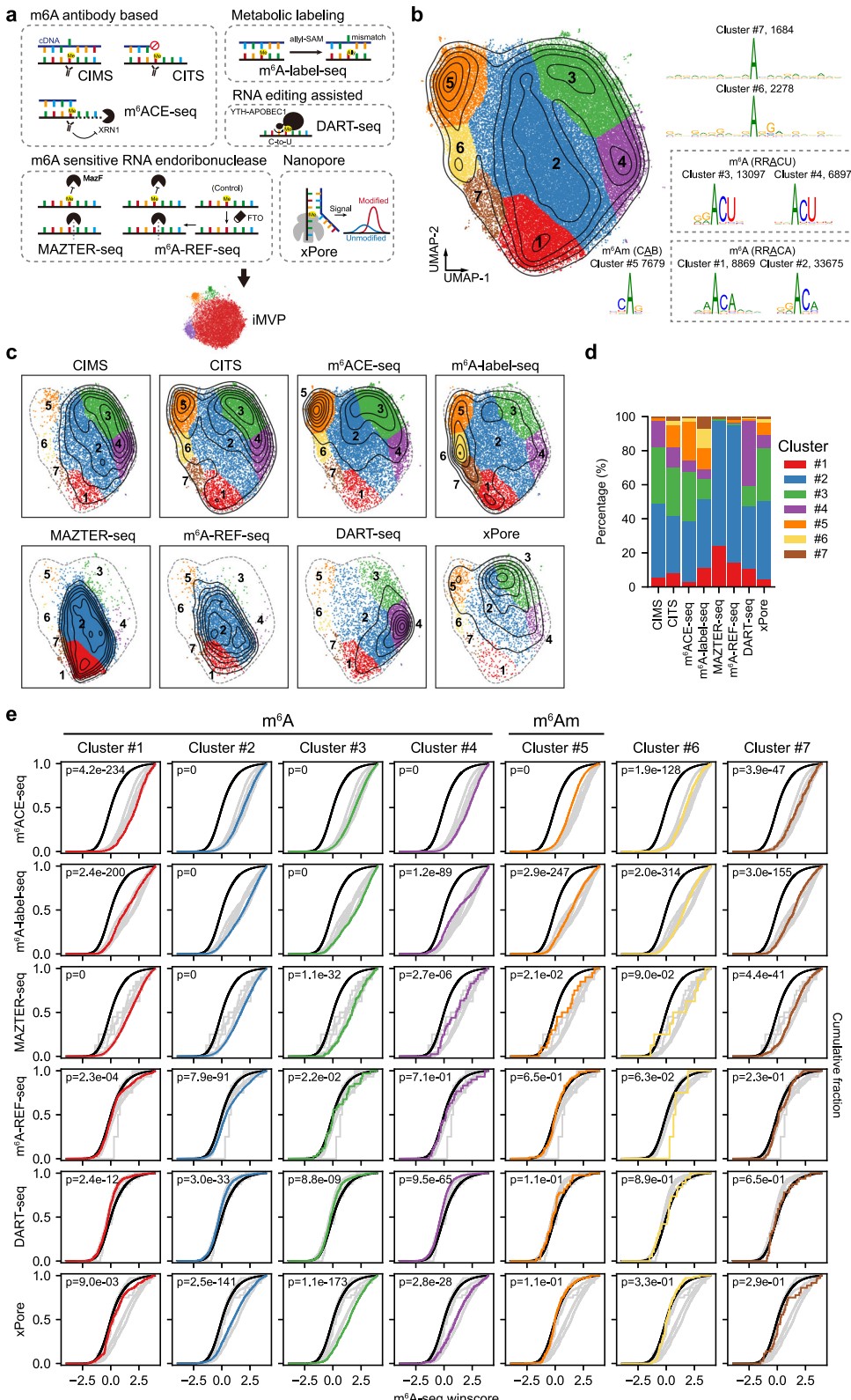

k-mer patterns, although repetitive non-Alu sites had a broader range of k-mer distribution compared with nonrepetitive sites. In contrast, Alu sites gathered into small clusters far away from the centroid of the projections.

We then separately applied iMVP to the three types of A-to-I editing sites to achieve a high-resolution view. We found a consistent "galaxy-like" sparse view of Alu sites, a "mitosis-like" two-centroid k-

mer distribution in repetitive non-Alu sites, and an "oocyte-like" pattern in nonrepetitive sites with a single centroid (Fig. 7b). To examine the clusters containing large numbers of data points in-depth, we applied an approximate clustering strategy based on a density histogram that mimicked the density sketching step in HDBSCAN (Supplementary Fig. 17a). With this strategy, we extracted 143, 52, and 22 large clusters from the projections. (Fig. 7c–e, Supplementary

**Fig. 6 | Comparison of eight methods of single-base resolution m⁶A/m⁶Am profiling with iMVP. a** The diagram showing the m⁶A profiling methods analyzed. CIMS and CITS, based on m⁶A antibody crosslinking to induce mutation (CIMS) or truncation (CITS) signatures during reverse transcription (RT); m⁶ACE-seq, m⁶A antibody crosslinking and followed by exonuclease digestion; m⁶A-label-seq, based on metabolic labeling of m⁶A to introduce mutation at m⁶A sites during RT; MAZ-TER-seq, based on the m⁶A-sensitive RNA endoribonuclease recognizing ACA motif; m⁶A-REF-seq, coupled the m⁶A-sensitive RNA endoribonuclease cleavage with FTO treatment as a control; xPore, based on the differential signals from m⁶A modified and unmodified bases in Nanopore direct RNA-seq; DART-seq, based on YTH-APOBEC1 fusion protein to introduce C-to-U mutation at sites adjacent to m⁶A

modification. **b** The global visualization of the aggregated 74,179 m⁶A candidates. The motifs and site numbers of the seven clusters were shown on the right. Cluster #1-4 are the canonical m⁶A RRACH motifs and cluster #5 is the canonical m⁶Am CAG motif. **c** The global visualization of m⁶A/m⁶Am profiles in different methods based on the result generated from the aggregated sites. **d** The percentages of sites called from different methods belonging to each cluster shown in **c**. **e** Validation of each cluster derived from antibody-independent profiling methods by m⁶A-seq enrichment scores. In each cumulative distribution plot, the cluster to be compared was highlighted in color, input windows were in black, and other clusters were in light gray. The *P* values were determined using a two-sided Kolmogorov-Smirnov test.

---

Fig. 17b–d, and Supplementary Fig. 18 and19). Interestingly, we found that the two centroids of repetitive non-Alu sites were located in poly(U)-tract and poly(A)-tract, respectively (Fig. 7d), which were mainly from LINE and LTR repeats (Fig. 7f). As homopolymer regions are prone to high sequencing error rates, we further validated these two groups of sites using ADAR1 knockout HEK293 cells[59]. We found that both sets of sites had decreased editing levels (Fig. 7g), confirming that these sites were authentic. For nonrepetitive sites, the centroid cluster was constituted by the canonical Position −1 G-depletion motif (Fig. 7e). Taken together, these analyses indicate that iMVP can handle extremely large datasets and expand our understanding of the sequence features of A-to-I RNA editing sites.

## Discussion

To reveal the biological meanings of RNA modifications, it is vital to identify authentic modification sites and correlate each of them to the corresponding writers. However, tools for addressing this issue are lacking. To solve this problem, we developed an exploratory data analysis workflow, iMVP, to subtype and visualize the compositions of putative RNA modification sites. With the help of dimension reduction and density-based clustering, we converted the motif discovery workflow, which traditionally occurs in a black box, into a transparent visualization issue. We systematically and quantitatively benchmarked various decomposition and cluster algorithms and selected the best combination for use in our framework development. Moreover, we developed an easy to use interactive version of our tool, which supports both automatic subtype clustering and intuitive subtype partitioning. Notably, although iMVP is not primarily designed for the search of gapped motifs, it is still possible to identify such motifs if the input dataset contains a sufficient number of sequences with gapped patterns. This is exemplified by the identification of two subtypes of Type III motifs that were found to be gapped motifs (Supplementary Fig. 13e, f). We demonstrated that iMVP analysis expands our understanding of the motifs and writers of RNA modifications and pinpoints noise that was previously difficult to identify. With the aid of GPU acceleration, iMVP is able to handle millions of sites that were previously unmanageable.

It is worthy to mention that the combination of UMAP and HDBSCAN is one of the solutions for iMVP. For decomposition, t-SNE also has excellent performance. For clustering, Leiden and Louvain may outperform HDBSCAN when the background is not noisy. Moreover, although 2-D spaces are used in both decomposition and clustering in this study, for some complicated datasets, it is possible to project the k-mer patterns onto a 3 or higher dimension space for further clustering.

It should be noted that iMVP has some limitations. First, iMVP is not suitable for application to datasets containing too few sites or sites without sequence preferences. Second, iMVP is a position-sensitive strategy, and even with a phase-matching strategy, the current version may have poor performance when dealing with datasets with high numbers of phases. Future improvements, such as the use of parametric UMAP[60] for dimension reduction, may improve its performance

on such datasets and broaden the usage of iMVP (e.g., global analysis in iCLIP datasets of RNA binding proteins). Third, in the clustering step, the cluster output may vary with different parameters, especially when extracting clusters from the condensed density tree. Although this issue may be partially solved by introducing a hyperparameter optimization strategy with DBCV scores[61], prior knowledge is still required for the selection of the proper parameters.

Given the recent rapid progress in the use of Nanopore direct RNA-seq to discover RNA modifications, it is expected that more tools will be developed to compare native and unmodified RNAs and identify differential sites, i.e., putative modification sites deposited by various writers. Hundreds of thousands to millions of sites may be found in a single Nanopore run. As we have shown that iMVP can handle the extremely large number of A-to-G variant sites, we expect iMVP to greatly aid in the visualization and interpretation of the Nanopore direct RNA-seq data in future studies.

Taken together, we anticipate that iMVP will be a valuable tool that can be rapidly adopted by the epitranscriptomics community and facilitate epitranscriptomic research in the future.

## Methods
### Cell culture
HeLa cells were purchased from Cell Bank, Type Culture Collection, Chinese Academy of Sciences (CBTCCCAS). HeLa cells have been identity verified using short tandem repeat (STR) analysis by CBTCCCAS. Cells were maintained in DMEM (Gibco, 11965118) supplemented with 10% FBS (CLARK, FB15015). Cells have been checked for mycoplasma contamination by CBTCCCAS and are routinely tested for mycoplasma by PCR detection of conditioned medium.

For Nocodazole treatment, wild type or knockout HeLa cells were first seeded and grown to 50% confluency. For Nop2 knockdown cells, cells were treated with siRNA for 48 hours before nocodazole treatment. Otherwise, cells were treated with 0.1 μg/ml nocodazole (Sigma, M1404) for 48 hours and then collected for further experiments.

### CRISPR/Cas9-induced mutagenesis
NSUN5 (gRNA, GTATGAGTTGTTGTTGGGAA) knockout cells were generated via CRISPR/Cas9-induced mutagenesis[19]. In brief, a gRNA sequence was designed using CRISPR-ERA (http://CRISPR-ERA.stanford.edu). The sgRNA template oligonucleotide was synthesized and cloned into lentiCRISPR v.2 plasmid (Addgene no. 52961). The plasmid was transfected into the cells using Lipofectamine 3000 (Thermo, L3000015) following the manufacturer's instructions. Transfected cells were selected using puromycin. Western blot with NSUN5 antibody (Proteintech, 15449-1-AP) was used to verify the loss of NSUN5 protein. NSUN2 and NSUN6 knockout cells generated previously were used in this study[19,21].

### RNA interference
For NOP2 knockdown, NOP2 siRNAs (siNOP2-1, GGAGUUCUUA-GAAGCUAAU; siNOP2-2, GAUCCAGCCGUGAAGACUATT) were transfected using Lipofectamine RNAiMAX (Thermo, 13778150)

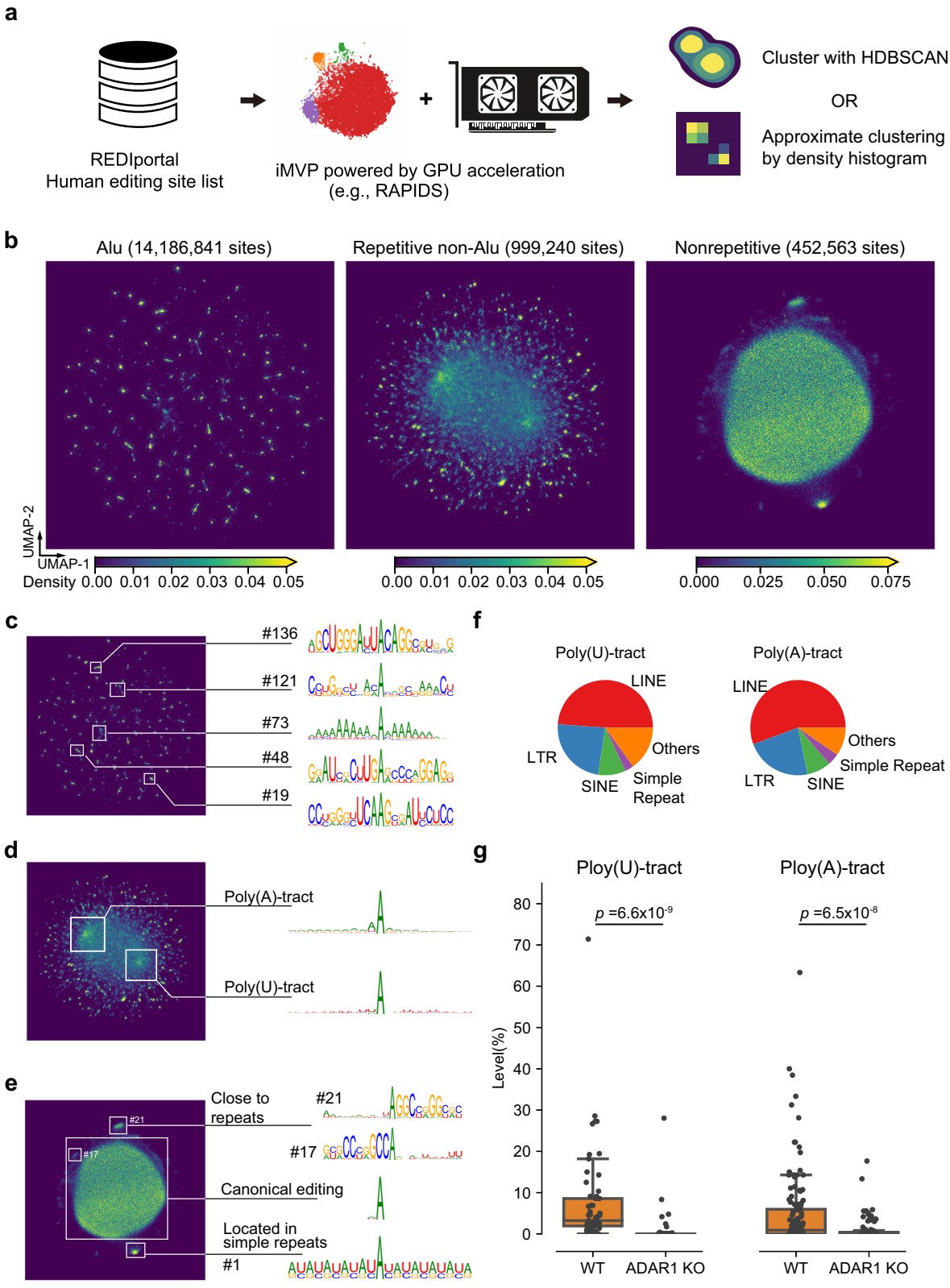

following the manufacturer's instruction. 48 hours after the transfection, the knockdown efficiency was examined by RT-qPCR with NOP2 primers (NOP2-F, GCACCCCAGGAACATGAG; NOP2-R, CACCACCTTGGGGAACTG) and GAPDH (as control) primers (GAPDH-F, TCAAGGCTGAGAACGGGAAG; GAPDH-R, GGACTCCAC GACGTACTCAG).

**mRNA BS-seq**

mRNA BS-seq library construction was performed[19]. In brief, total RNA was isolated with TRIzol reagent and Direct-zol RNA MiniPrep kit. poly(A) + RNA was separated from total RNA using Oligo dT Magnetic Beads (Vazyme, N403). 100 ng−1 μg of poly(A) + RNA was converted using the EZ RNA methylation kit (Zymo Research, R5002) with a

**Fig. 7 | GPU-accelerated iMVP analysis of an extremely large A-to-I RNA editing dataset in REDIportal. a** The schema of the analysis. The dimension reduction was done with UMAP API in RAPIDS. The large clusters in the projections were then extracted by HDBSCAN or density histogram. **b** The density histogram of UMAP projections of 21-mers of A-to-I editing sites in Alu, repetitive non-Alu, and non-repetitive regions. 3,845,894, 794,349, and 441,674 different types of k-mers were analyzed separately. Then the real number of sites was used for density calculation. **c-e** Examples of the motifs of specific clusters from Alu, repetitive non-Alu, and non-repetitive sites. **f** The fractions of different types of repeat elements in the poly(U)-tract and poly(A) tract shown in **d**. **g** Boxplots showing the editing level changes of editing sites located in the Ploy(U)-track and Ploy(A)-tract between wild-type (WT) and ADAR1 knockout (KO) HEK293 cells. n = 1 for each type of cells. We required that the sites were covered by at least 10 reads in both samples. *P* values were calculated by a two-sided paired samples Wilcoxon test. Boxplots: 25th to 75th percentiles (boxes), medians (horizontal lines), and 1.5 times of the interquartile range (whiskers).

modified high-stringency conversion condition (Sulfonation: 3 cycles, (1) 70 °C, 10 min; (2) 64 °C, 45 min; Desulfonation: 25 °C for 30 min). The converted RNA was fragmented into 150 – 200 nt fragments in fragmentation buffer (NEB, E6150) at 94 °C for 8 min. The fragmented RNA was then used for library construction using NEBNext Ultra II Directional RNA Library Prep Kit. Libraries were sequenced on Hiseq X10 (Illumina) to produce paired-end 150 bp reads (Supplementary Data 6).

### m⁶A-seq
Polyadenylated RNA, separated from total RNA using Oligo dT Magnetic Beads (Vazyme, N403), was fragmented in 1X NEB Next Magnesium RNA Fragmentation Buffer at 94 °C for 5 min. 10 ng of fragmented RNA was saved as input. 10 μg fragmented RNA was further incubated with 5 μg rabbit anti-m⁶A polyclonal antibody (Synaptic Systems, catalog number 202003) in IPP buffer (10 mM Tris-HCl pH 7.4, 150 mM NaCl and 0.1% Igepal CA-630) overnight at 4 °C. The m⁶A-Ab mixture was then immunoprecipitated by incubation with protein-G magnetic beads at 4 °C for another 2 hours. RNA was then eluted from the beads with IPP buffer containing 0.5 mg/ml N6-methyladenosine (Sigma-Aldrich, M2780). VAHTS stranded mRNA-seq library prep kit (Vazyme, NR601) was used for library construction. Libraries were sequenced on HiSeq X (Illumina) to produce paired-end 150 bp reads (Supplementary Data 6).

### Data pre-processing
A summary of the source of data is in Supplementary Data 6.

**m⁵C analysis.** All m⁵C sites analyzed were obtained with standard BS-seq library preparation and analysis workflow[19,32]. These m⁵C sites were in GRCh37 or GRCm38 coordinates.

**xPore analysis.** The processed data of HEK293T (in GRCh38 coordinates) were obtained from the supplementary table of xPore[26]. We retrieved the strand information of the sites based on the provided gene ID.

**m⁶ACE-seq data analysis.** We downloaded the site list from the supplementary table of Koh et al[51].

**ModTect analysis.** The site lists of ModTect in hg19 (GRCh37) coordinates were obtained from the supplementary table. Only sites with no ambiguous strand direction were used in our analysis. We spiked 14 known modified sequences (Supplementary Data 5) into the ModTect dataset for analysis.

**m⁶A method comparison.** We performed the m⁶A method comparison analyses using Ensembl GRCh38 coordinates. We first downloaded the site list from the supplementary tables of Linder et al[10]. (CITS/CIMS), Shu et al[55]. (m⁶A-label-seq), Pandey et al[56]. (MAZTER-seq), Zhang et al[57]. (m⁶A-REF-seq), Meyer[58] (DART-seq), and Pratanwanich et al[26]. (xPore). For m⁶A-label-seq data, we retrieved the genomic coordinates based on the RefSeq id and sequences. For DART-seq, we assigned the m⁶A sites adjacent to the C-to-U signals. For sites in hg19

format, we used UCSC liftOver tool to convert them to Ensembl GRCh38 format. The summarized site list is in Supplementary Data 2.

**RNA editing analysis.** The site list (hg38) was downloaded from REDIportal[25]. We grouped the sites into Alu, repetitive non-Alu, and nonrepetitive sites based on the annotations in the original table.

### The iMVP framework
For each analysis, we provided the codes, example inputs, and example outputs in detail in Juypter-Notebook (https://github.com/sysu-zhanglab/iMVP). Our package (including helper functions and the interactive interface) is available on Python Package Index (Pypi): iMVP-utils. Documents of our analysis and the package can be found on https://imvp.readthedocs.io/. In brief, we encoded the 10 nt flanking sequences of the sites into one-hot encoded format, and then projected the encoded sequences onto a 2-D plane with UMAP. We used HDBSCAN to split the projections based on the density contours. If stubborn clusters (e.g., two high-density clusters connect with a linkage) encountered, we further split those clusters with HDBSCAN and merged the small clusters manually based on the density contours. Density Based Cluster Validity (DBCV), which is implemented in the HDBSCAN API, may be used to search for suitable combination of the parameters. For Leiden, the Significance of Hierarchical Clustering (SHC) may be used to calculate the quality of clustering[62].

In particular, for phase matching, we first built the 21-mer sequences for the −2, −1, 0, +1, +2 phases of the signals, then we performed phase-matching based on the patterns of clusters. Coordinates of phase-matched sites were re-computed to match the positions of the "A"s. For RNA editing site analysis, we used a single RTX2080Ti GPU for UMAP analysis. In the test run, we found a 46-fold boost in UMAP, allowing us to project up to 5 million 21-mer sequences within 15 min. However, CPU-based HDBSCAN is still the best practice because of the shorter runtime and the more flexible memory allocation. The computation of UMAP of k-mers was finished within 1 hour, where the majority of the time was spent on I/O and encoding process rather than UMAP computation (less than 15 min).

We set up our analysis (CPU-based) with Python 3.7, and the following core packages were used: Pandas (v1.3.4), Numpy (v1.20.0), Scipy (v1.5.1), Scikit-learn (v0.23.1), biopython (v1.77), hdbscan (v0.8.27), umap-learn (v0.5.2), openTSEN (v0.6.1), louvain (v0.7.1), leidenalg (v0.8.8), dash (v2.2.0), dahs-bio (v0.9.0), imageio (v2.13.5), weblogo (v3.7.0), and opencv-python (v4.5.5). For GPU-based analysis, we used the Docker container provided by NVIDIA: RAPIDS release 22.02 in Ubuntu 18.04, Python 3.8, and CUDA 11.5. The workflow was run on a RTX2080Ti with 11 Gb memory. The parameters and operations of each analysis were summarized in Supplementary Data 7.

### Calculation of outgroup-ingroup score and boundary score
Outgroup-ingroup score was defined as the coefficient of variation (CV) ratio between outgroup distances and ingroup distances. Ingroup and outgroup distances were defined as the Euclidean distances between a data point to the centroid of a group: $D = \sqrt{(x - \bar{x})^2 + (y - \bar{y})^2}$, where $x$ and $y$ are the X-Y coordinate of a data

point, and $\bar{x}$ and $\bar{y}$ are the mean of all data points of a group. Then the standard deviation of ingroup distances (SDingroup) and outgroup distances (SDoutgroup) and the means of them (Meaningroup and Meanoutgroup) were calculated. Outgroup-ingroup score was then obtained via the following formula: Score $= \frac{CV_{outgroup}}{CV_{ingroup}} = \frac{SD_{outgroup}/Mean_{outgroup}}{SD_{ingroup}/Mean_{ingroup}}$. If sites in a group are more condensed, a lower ingroup CV is expected; if the sites in different groups are better separated, a higher outgroup CV is expected. Hence, algorithms with better performance will have higher outgroup-ingroup scores.

Boundary score was calculated as below. We first searched for the top 50 nearest neighbors for each data point. Next, we calculated the proportions (*P*) of the neighbors that were not in the same group of the data point. Last, all proportions were aggregated to draw a cumulative distribution (Supplementary Fig. 1a). Boundary score was defined as the ratio of the sites whose $P > 0.1$. Hence, higher boundary scores mean that more sites were projected to form ambiguous boundaries. This score is complementary to the outgroup-ingroup score to describe the discernibility of the algorithm.

### Calculation of ARI
ARI was calculated by sklearn.metrics.adjusted_rand_score function.

### Parameter optimization for MEME, STREME and HOMER
For MEME, we fine-tuned parameters including maxw, mods, markov_order, and objfuns. Given that our simulation dataset comprised 10 motifs, each with a length exceeding 6, we established specific parameter values to optimize the performance of MEME. We set the minimum motif width (minw) to 6 and the number of motifs (nmotifs) to 10. Additionally, we explored various values for the maximum motif width (maxw), including 8, 12, and 15. We also varied the Markov model orders (markov_order) from 0 to 3. The mod parameter encompassed zoops, anr, and oops schemas, while the objfuns parameter encompassed the classic, de, se, cd, nc, and ce algorithms. This exploration resulted in the generation of 192 parameter sets for comprehensive testing.

For HOMER, to address the constraints of HOMER, which is limited to searching for motifs of a single specific length, we performed an analysis using a set of seven length parameters, spanning from 6 to 12. We also set the number of motifs to 10. The mean values were used to represent the performance of HOMER.

For STREME, we adjusted the two parameters: maxw and objfun. The maxw parameter was varied across three values: 8, 12, and 15. while the objfun parameter included two options: cd or ce. This led to 6 parameter sets being tested.

### E-value calculation
E-values of the motifs were calculated by Simple Enrichment Analysis (SEA) of MEME (v5.0.0). All *E*-values, *p*-values, and *q*-values of the motifs presented in the study were shown in Supplementary Data 8.

### KL divergence
KL divergences were calculated by MotifSuite[63]. A customized script based on Biopython.motifs API was used to generate the PWM format input for MotifSuite.

### Motif drawing and motif finder analysis
We used Weblogo (v3.7.0) to plot the motifs of selected sequences. MEME/ STREME (v5.3.3) and HOMER (v4.11) were used in motif discovery.

### mRNA BS-seq data analysis
mRNA m[5]C sites were called as we previously described[19]. We first trimmed adapters, the first 10 bp of the reads, the last 6 bp of the reads, and the low-quality bases using Cutadapt (-e 0.25 -q 25 -trim-n)[64]

and Trimmomatic[65]. Then clean reads were mapped to the in silico converted genome by HISAT2 (-k 10,−fr,−rna-strandness FR,−nomixed)[66] to obtain unique alignments. The remaining unmapped and multiple mapped reads were further mapped to the in silico converted transcriptome by Bowtie2 (-end-to-end,−fr,−gbar 5,−mp 5, -k 10, -R 2, -D 5)[67]. Alignment results were merged, and only bases with high quality ($Q \geq 30$) were used for the variant calling. Last, the sites were called using a series of filters as previously described[19]. The methylation level (mismatch frequency) is defined as the number of reads with C divided by the number of reads with C or T.

### RNA structure prediction and illustration
RNA secondary structure was predicted using RNAfold (2.4.12)[68] with default settings. The m[5]C sites and 50 nt flanking sequences were extracted from the genome. The folding frequencies of each base of the sites with their 25 nt flanking sequences were shown.

### m[6]A-seq analysis
We first trimmed adapters using Cutadapt[64] (-e 0.1 -q 20 -m 20 --trim-n). Cleaned reads were then mapped to GRCh38 (Ensembl release 104) genome and transcriptome by HISAT2 with default settings in strand-specific mode[66]. Only unique alignments were transformed into bedGraph for further analysis. m[6]A peak analysis was performed as previously described with some modification[50]. In brief, each gene was split into 50-nucleotide sliding windows and an enrichment fold (winscore) was calculated for each window: winscore = $\log 2(\frac{MeanWinIP/MedianGeneIP}{MeanWinControl/MedianGeneControl})$. MeanWinIP and MeanWinControl are the mean coverage for each window for immunoprecipitation and input control, respectively. MedianGeneIP and MedianGeneControl are gene median coverages for immunoprecipitation and input control, respectively. To examine the enrichments of each cluster, we compared the cumulative distributions of the winscores of each cluster with background winscores.

### Reporting summary
Further information on research design is available in the Nature Portfolio Reporting Summary linked to this article.

## Data availability
The data supporting the findings of this study are available from the corresponding authors upon reasonable request. The sequencing data generated in this study have been deposited in the GEO database: BS-seq of HeLa cells with NSUN protein deficiency: GSE197650; control HeLa cells: GSM5319029; m[6]A-seq data generated in this study: GSE198955. Source data for the figures and supplementary figures are provided as a Source Data file. Source data are provided in this paper.

## Code availability
The source code is published at https://github.com/SYSU-zhanglab/iMVP. https://doi.org/10.5281/zenodo.8286943.

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

## Acknowledgements
This study was supported by grants from National Key R&D Program of China (2020YFA0509400 to R.Z.), Guangdong Innovative and Entrepreneurial Research Team Program (2016ZT06S638 to R.Z.), the National Natural Science Foundation of China (82173050 to T.H.), Guangdong Natural Science Foundation (2021A1515010667 to T.H.), and SUMC Scientific Research Initiation Grant (510858061 to T.H.).

## Author contributions
J.L. and R.Z. contributed to the study design. J.L. and J.Y. performed bioinformatics analysis. T.H., T.Z., and Y.Z. performed all experiments. J.L. and R.Z. wrote the manuscript with input from all authors.

## Competing interests
The authors declare no competing interests.
