## [Peer Review File · Nature Communications]

Epitranscriptomic subtyping, visualization, and denoising by global motif visualizationREVIEWER COMMENTS

Reviewer #1 (Remarks to the Author):

Liu and colleagues present their novel method for finding sequence motifs among thousands of nucleic acid sequences and show how it can be used to find the target specificity of RNA modifications. Their new algorithm, iMVP, deftly applies large-scale dimension reduction techniques to reveal similarity relationships among sequences provided. iMVP can handle tens of thousands of sequences with reasonable computational costs while revealing clusters of minor motifs because of the scalability and effectiveness of the dimension reduction algorithms. They open up new opportunities for regulatory motif research with their interesting approach.

Despite its novelty and evidence of useful applications, this manuscript requires significant improvements in the evaluation and benchmarks of the algorithmic performance. The algorithm and software are the manuscript's main focus because the scientific discoveries described in it are confirmations rather than brand-new discoveries. Consequently, this necessitates a more stringent evaluation of algorithmic characteristics, as described below.

1. The cases presented in the paper appear too homogeneous to accurately illustrate the algorithm's characteristics. Simulations with synthetic datasets would be required to evaluate the more diverse cases, such as starting with >100k or 100 sequences, dealing with >10 heterogeneous motif clusters, and dealing with highly noisy circumstances.
2. The authors sought rigorously optimal parameters and configurations for the evaluation but put too little effort into optimizing MEME or HOMER running. There is significant room for improvement in the utilization of the existing tools.
3. The authors claim that iMVP performs better than the current motif finders when working with many sequences, so they must carefully assess the algorithm's scalability. For >100k sequences, I'm not sure if iMVP would require less computation time and memory than MEME or CD-HIT.
4. Although iMVP is not intended for gapped motifs, if the input has enough sequence examples, it may be able to tolerate small gaps. It would be great if this possibility were investigated or discussed.
5. Because of its position dependency, iMVP can only be used in a limited range of applications where single-nucleotide resolution detection is feasible. Most motif-finding algorithms permit sequences to have a relaxed appearance of position. The iMVP algorithm should become more interesting when extended to this relaxation.

Reviewer #2 (Remarks to the Author):

This manuscript provides a simple but effective way to visualize the RNA modification into 2D dimension visualization space. The input k-mer sequences were encoded through one-hot encode, and define a Euclidean distance among different sequence, UMAP were applied to project sequence into 2D space, and used HDBSCAN to cluster sequences. The authors also provide some realistic scenarios to prove the capability of their methods on visualization and interpretation of epitranscriptomic data. In general, I only have the following 5 questions or concerns that need authors to reply or polish their manuscripts through some extra analyses.

Q1: It's dangerous to over-interpret the coordinate of non-linear dimensional reduction UMAP or t-SNE. The ability of capture global data structure of both methods highly depend on the initialization (<https://www.nature.com/articles/s41587-020-00809-z#ref-CR2>). There are also some debates on if there exists extensive distortions on 2D embedding space (<https://www.biorxiv.org/content/10.1101/2021.08.25.457696v4.full.pdf>). Using distance defined in 2D space as the metric may not be sufficient and even could be wrong.

Q2: Following Q1, I think the key goal (correct me if I am not right) in the first step is subtyping. Giving a good representation on 2D space is not the ultimate goal. But finding latent subtypes of sequence (modification) is the goal. It is also the same thing in cell type identification of single cell analysis. Therefore in single cell gene expression analysis, the normal routine of clustering is to first use Top-K principal components (PCs) to build neighborhood graphs and use graph-based methods, e.g. leiden and louvain to cluster cells. A DBSCAN based clustering on t-SNE coordinate is used in a very old version of Seurat (V2 I think), but has been discarded. So I am wondering how the author performs louvain and leiden clustering (I didn't find this part in their github repo <https://github.com/SYSU-zhanglab/iMVP/tree/main/Benchmarking>). The author needs to show that using the UMAP+HDBSCAN method is indeed better than KNN graph+leiden or louvain clustering. In general, I hope the author gives a more rigorous benchmark to support that UMAP+DBSCAN is the best clustering algorithm. But downstream data visualization could be UMAP or t-SNE, I merely question about their clustering algorithm.

To benchmark the clustering performance, one could use other metrics specifically design for clustering analysis, e.g. Adjusted Rand Index (ARI). To compare your unsupervised learned label and ground truth label (Type I and Type II).

Q3: The motifs type divergence across species need to be clearer. From Fig.3g, it's hard to tell the motif preference divergence between fly and vertebrate Type I site. I suggested the author need to fully describe the motif preference in vertebrate type I site, to make comparison with fly. Or use some statistical metrics, e.g. KL divergence, to illustrate the difference in motif state distribution.

Q4: The authors claim that clustering motifs to have global motif visualization of the inputs may facilitate the identification of false positives. In general, it is, however, in the main text, those false positives belong to the known sequence error or artifacts, but in some cases without prior knowledge, how do we know a new sequence feature is a biological meaningful "feature" but not artifacts?

Q5: Following Q4, What's more, in the clustering stage, some newly identified clusters may not be meaningful, and clustering algorithms (including HDBSCAN) will partition data even in cases where there is only uninteresting random variation present. Therefore, in clustering there also exists a risk of introducing new artifacts, so assessing how much the clustering results can change due to uninteresting random variation is very important. The author needs to provide some methods to examine the statistical significance of each cluster, or statistical possibility of under- or over-clustering (<https://www.biorxiv.org/content/10.1101/2022.08.01.502383v1.full.pdf>). That would make the results more reliable but not highly dependent on how much we know about biology.

Reviewer #3 (Remarks to the Author):

The authors describe a new approach to analyze motifs in base modification sequencing data using 2D projection and clustering of one-hot encoded 21-mers. The method is an interesting application of dimensionality reduction and clustering, and the authors demonstrate its potential utility for data exploration. The most convincing finding points to NSUN5 and Nop2 modification of mRNA sites. However, a number of other claims are overstated or require further validation.

Major comments

1. Direct RNA nanopore sequencing is error prone. What are current estimated rates for base calling errors and to what extent would this affect the data included in these analyses? The authors state, for example, "Other small clusters were centered by non-A bases (Extended Data Fig. 3b-e) and were less likely to be m6A signals; thus, they were not included in the analysis." (p 7).
2. m6Am primarily follows 7-methylguanosine caps at the 5' ends of mRNA (see Mauer et al., 2016; Boulias et al., 2019; etc.). When the authors claim that "The remaining 5% of sites were within the CAG motif, which is an m6Am-like motif" (p 7) -- 1) do they have a citation for m6Am in CAG motifs? 2) What fraction of this cluster of sites was at the 5' end of mRNA? Does further exploring the hypothesis that these sites are m6Am modified make sense given their location or could these sites be ruled out as true m6Am sites based on non-5' location alone?
3. "Taken together, iMVP assists in the identification of a previously unknown trans- regulatory evolution between vertebrate and fly NSUN2 proteins" (p 10)
This seems like unnecessary jargon to describe results suggesting that homologous enzymes from different phyla have slightly different motif preferences. What is "trans-regulatory evolution"? Have the authors verified that the differences they see are not due to differences in the prevalence of the motifs in each species' transcriptome rather than differences in the specificities of the enzymes?
4. "These observations highlight the ability of iMVP to pinpoint false positives that were unaddressed by traditional filtering steps." (p 10)

What were filtering parameters and how many true sites would have been lost if filtering was adjusted to exclude these sites? Can the authors include a ROC plot or report area under the curve depending on filter threshold?

5. “Interestingly, we found that the two centroids of repetitive non-Alu sites were located in poly(U)-tract and poly(A)-tract, respectively (Fig. 7d), which were mainly from LINE and LTR repeats (Fig. 7f). For nonrepetitive sites, the centroid cluster was constituted by the canonical Position -1 G-depletion motif (Fig. 7e). Taken together, these analyses indicate that iMVP can handle extremely large datasets and expand our understanding of the sequence features of A-to-I RNA editing sites.” (p 15)

How are the authors confident that these sites are true positives?

6. “the cluster output may vary with different parameters, especially when extracting clusters from the condensed density tree.”

To what extent does the cluster output vary? Also, UMAPs are based on a stochastic algorithm. Between this and sensitivity to parameter selection, how reproducible are results?

7. Why did the authors choose to encode 21-mers rather than another length of motif and does this choice affect iMVP performance?

Minor comments

1. In the introduction, the authors state, “writers of m6A and A-to-I editing are misregulated in multiple types of cancers and have been identified as promising therapeutic targets of cancers, with related cancer treatments under development”. Can they expand or offer citations on which cancer treatments are actually under development targeting m6A or A-to-I editing writers and their stage of development? Briefly consulting a 2021 review from Yang et al., (“RNA methylation and cancer treatment” <https://doi.org/10.1016/j.phrs.2021.105937>) does not identify any therapies targeting writers approaching clinical trials.

2. “the similarities of the sequences were therefore represented by the Euclidean distances in the projections” implies that these distances are accurate representations of similarity. However, see for instance Chari and Pachter (bioRxiv, 2022) and their references, showing that distances are inevitably distorted with dimensionality reduction of many points. I would suggest changing “represented” to “approximated”.

3. “also true for more complicated m5C data from nocodazole (Noc) treated HeLa cells” (p. 5) For readability, please introduce here the expected or measured effects of nocodazole on m5C. How is this data more complicated?

4. “Globally, we found that only a small set of m6A and m6Am sites were overlapped among methods (Extend Data Fig. 7a), although the same cell type was used in all studies. This observation suggests that each method only captured a subset of the methylated sites.”

Alternatively, each method captures many false positives or m6A levels vary in this cell type. Do FP rates differ among these methods? Which cell type was used to generate these datasets?

5. "Compared with MAZTER-seq, signals from m6A-REF-seq had much lower m6A-peak enrichment (Fig. 6e), suggesting that the addition of FTO treatment might introduce noise."

A table in the main text describing the differences among these methods would be helpful.

We thank the reviewers for examining our manuscript and providing constructive feedback. We performed additional analyses and experiments to address questions raised by the reviewers. The changes are highlighted in **Yellow** in the revised manuscript for easy tracking. Please see below for our point-by-point response to reviewers' comments.

Reviewer #1 (Remarks to the Author):

Liu and colleagues present their novel method for finding sequence motifs among thousands of nucleic acid sequences and show how it can be used to find the target specificity of RNA modifications. Their new algorithm, iMVP, deftly applies large-scale dimension reduction techniques to reveal similarity relationships among sequences provided. iMVP can handle tens of thousands of sequences with reasonable computational costs while revealing clusters of minor motifs because of the scalability and effectiveness of the dimension reduction algorithms. They open up new opportunities for regulatory motif research with their interesting approach.

Despite its novelty and evidence of useful applications, this manuscript requires significant improvements in the evaluation and benchmarks of the algorithmic performance. The algorithm and software are the manuscript's main focus because the scientific discoveries described in it are confirmations rather than brand-new discoveries. Consequently, this necessitates a more stringent evaluation of algorithmic characteristics, as described below.

1. The cases presented in the paper appear too homogeneous to accurately illustrate the algorithm's characteristics. Simulations with synthetic datasets would be required to evaluate the more diverse cases, such as starting with >100k or 100 sequences, dealing with >10 heterogeneous motif clusters, and dealing with highly noisy circumstances.

We thank the reviewer for this suggestion. To test iMVP in more diverse situations, we simulated two datasets.

The first dataset (marked as the large simulation dataset) contains 12 randomly chosen transcription factor motifs from JASPER. The lengths of the motifs vary from 6 bp to 15 bp, and the numbers of motifs vary from 100 to 20,000. We also include 50,000 random noises. In total, this dataset has 109,850 different sequences. The ground-truth was shown in **Extended Data Fig. 4a and Supplementary Fig. 1**. iMVP and MEME identified the enrichment of 9 and 8 of the 12 motifs, respectively (**Extended Data Fig. 4**). One of the least abundant motifs (JUNB) containing 250 records (1.8% of the dataset) can be only found by iMVP. Notably, with MEME, we had to specify the number of motifs to search for based on our prior knowledge (in this case, we set it to 12). However, with iMVP, no prior knowledge of motif number was required.

The second dataset (marked as the small simulation dataset) contains 5 motifs. The lengths of the motifs vary from 6 bp to 15 bp, and the numbers of motifs vary from 5 to 50. We also include 100 random noises. In total, this dataset has 200 different sequences (**Supplementary Fig. 2**). For this dataset, both iMVP and MEME found the major motifs (**Extended Data Fig. 4**).

We have included these results in the main text (**Page 7, Lines 174-180**).

“To compare the performance of iMVP with other traditional tools in motif search, we analyzed two simulation datasets that comprised 100,000 and 200 sequences with 12 and 5 heterogeneous motifs, respectively (Supplementary Note 1). Our initial comparison was between iMVP and MEME. In the larger dataset, iMVP demonstrated superior ability to identify less abundant motifs compared to MEME, whereas both methods performed similarly in the smaller dataset (Extended Data Fig. 4, Supplementary Fig. 1-2 and Supplementary Note 3).”

Extended Data Fig. 4. Comparison of iMVP with MEME.

(a) UMAP projection of the large simulation dataset (ground-truth). The simulated motifs were shown in different colors, random noises were shown in gray.

(b) iMVP analysis of the simulated data in (a). The details of enriched motifs were shown in Supplementary Fig. 1.

- (c-d) UMAP projection and motifs of the small simulation dataset.
 (e-f) iMVP analysis and motifs for the small simulation dataset.
 (g) MEME output for the small simulation dataset.

TF	Number	Ground truth	iMVP	MEME
HAT 5	20000			
CEB PB	10000			
FOX A1	8000			
SOX 6	6000			
NFK B1	5000			
ELF1	4000			
YY1	3000			
ESR 2	2000			
PDX 1	1000		NA	NA
HOX A5	500		NA	NA
JUN B	250			NA
ZEB 1	100		NA	NA
Clust er #10 (iMVP)				

Supplementary Fig 2. Detailed information for the analyses on the large simulation dataset.

TF	Number	Ground truth	iMVP	MEME
CEBPB	50			
SOX6	25			
ESR2	10			

Supplementary Fig. 3. Detailed information for the analyses on the small simulation dataset.

2. The authors sought rigorously optimal parameters and configurations for the evaluation but put too little effort into optimizing MEME or HOMER running. There is significant room for improvement in the utilization of the existing tools.

We thank the reviewer for this comment. To address this question, we optimized the running of traditional tools using the large simulation dataset mentioned above.

For MEME, we fine-tuned parameters including maxw, mods, markov_order, and objfun. Given that our simulation dataset comprised 10 motifs, each with a length exceeding 6, we established specific parameter values to optimize the performance of MEME. We set the minimum motif width (minw) to 6 and the number of motifs (nmotifs) to 10. Additionally, we explored various values for the maximum motif width (maxw), including 8, 12, and 15. We also varied the Markov model orders (markov_order) from 0 to 3. The mod parameter encompassed zoops, anr, and oops schemas, while the objfun parameter encompassed the classic, de, se, cd, nc, and ce algorithms. This exploration resulted in the generation of 192 parameter sets for comprehensive testing.

For HOMER, to address the constraints of HOMER, which is limited to searching for motifs of a single specific length, we performed an analysis using a set of seven length parameters, spanning from 6 to 12. We also set the number of motifs to 10. The mean values were used to represent the performance of HOMER.

For STREME, we adjusted the two parameters: maxw and objfun. The maxw parameter was varied across three values: 8, 12, and 15. while the objfun parameter included two options: cd or ce. This led to 6 parameter sets being tested.

To evaluate the performance of different methods, we assessed the computational time and the accuracy of motif searching. Overall, we found that iMVP surpassed

existing tools in terms of both computational time and accuracy (**Extended data Fig. 5**). We have included this in the revised manuscript (**Page 7, Lines 182-188**).

Extended Data Fig. 5. Comparison between iMVP and other motif search tools.

(a) Comparison of computational time among different methods. Red star, iMVP; green square, the combined result of 7 HOMER parameters; yellow triangle, six parameters of STREME; blue circle, 192 MEME parameters.

(b) Comparison of memory consumptions among different methods.

(c) Number of motifs detected by different methods.

(d) Median cosine similarity between motifs discovered by different tools and ground-truth motif position weights.

(e) Number of true-positive motifs (defined as the ones with cosine similarity greater than 0.9) found by different tools.

(f) Accuracy, defined as the number of true-positive motifs divided by the number of all motifs discovered, of different methods.

3. The authors claim that iMVP performs better than the current motif finders when working with many sequences, so they must carefully assess the algorithm's

scalability. For >100k sequences, I'm not sure if iMVP would require less computation time and memory than MEME or CD-HIT.

We thank the reviewer for this comment. As we mentioned in our response to Question 2, we have conducted a comprehensive comparison between MEME and iMVP.

CD-HIT is a software program specifically designed for the purpose of clustering and comparing nucleotide sequences. Its primary function is to detect and eliminate redundancy within sequence datasets, particularly in large-scale collections such as transcriptomes or metagenomes. While CD-HIT is indeed useful for reducing sequence redundancy, it is not intended for motif discovery. We attempted to utilize CD-HIT for motif discovery by applying it to the large simulation dataset of 100k sequences. However, we observed that it reduced the redundancy of the sequences to approximately 50k. Since the information regarding sequence redundancy is crucial for our analysis aimed at identifying enriched patterns, we made the decision not to utilize CD-HIT for motif discovery in this context.

4. Although iMVP is not intended for gapped motifs, if the input has enough sequence examples, it may be able to tolerate small gaps. It would be great if this possibility were investigated or discussed.

We thank the reviewer for this comment. We agree with the reviewer that iMVP is capable of identifying gapped motifs with small gaps. For example, in **Extended Data Fig. 10e&f**, the two subtypes for Type III motifs methylated by NSUN5 were both gapped motifs. We have clarified this in the revised manuscript (**Pages 17-18, Lines 478-482**).

“Notably, although iMVP is not primarily designed for the search of gapped motifs, it is still possible to identify such motifs if the input dataset contains a sufficient number of sequences with gapped patterns. This is exemplified by the identification of two subtypes of Type III motifs that were found to be gapped motifs (Extended Data Fig. 10e&f).”

5. Because of its position dependency, iMVP can only be used in a limited range of applications where single-nucleotide resolution detection is feasible. Most motif-finding algorithms permit sequences to have a relaxed appearance of position. The iMVP algorithm should become more interesting when extended to this relaxation.

We thank the reviewer for this great suggestion. Following reviewer's suggestion, we introduced a sliding window strategy to broaden its applications. The sliding window strategy consists of three steps: (1) generating sliding windows on input sequences; (2) performing iMVP on all the windows to extract clusters; (3) conducting motif alignment and comparison on the extracted clusters to identify the patterns. To test this strategy, we utilized a simulation dataset comprising 500 PDX1 motifs randomly distributed in a 50-bp sequence, along with 500 random noise sequences. We first generated 20-bp windows with a 1-bp step, resulting in a total of 30,000 windows. Then we applied iMVP to these 30,000 windows (**Extended Data Fig. 8a&b**) and extracted enriched patterns (clusters #1 to #11) (**Extended Data Fig. 8c**). Among these patterns, 10 exhibited distinct PDX1 motifs. These motifs may be further

aligned with Tomtom to generate a consensus motif (**Extended Data Fig. 8d**). By following all three steps, we identified between 48 to 483 sequences containing PDX1 motifs. The false discovery rate (FDR) ranged from 7.7% to 45.5%, depending on the specific thresholds applied (**Extended Data Fig. 8e-g**). We have included the results in the revised manuscript (**Pages 9-10, Lines 237-251**).

Extended Data Fig. 8. Sliding window strategy of iMVP on a simulated dataset. (a) The ground-truth X-Y locations of the sliding windows generated from simulated PDX1 motif containing sequences. PDX1 containing sequences were enriched at the edge of the UMAP projections.

- (b) iMVP result for 30,000 sliding windows generated on 500 50 bp sequences containing PDX1 motif as well as 500 50 bp random noises.
- (c) Motifs discovered by iMVP in (b).
- (d) An example of motif alignment between cluster #1 and cluster #6 by Tomtom.
- (e-g) The metrics for the sliding window strategy of iMVP. A sequence was considered to contain a PDX1 motif if it satisfied the criterion of having a certain number of sliding windows belonging to clusters #1-9 and cluster #11, as determined by a predefined threshold. Then metrics were calculated based on true positives, true negatives, false positives, and false negatives defined by the threshold. Three different metrics were present: threshold vs false discovery rate (FDR) (e), number of false positives vs true positives (f) and precision vs recall (g).

Reviewer #2 (Remarks to the Author):

This manuscript provides a simple but effective way to visualize the RNA modification into 2D dimension visualization space. The input k-mer sequences were encoded through one-hot encode, and define a Euclidean distance among different sequence, UMAP were applied to project sequence into 2D space, and used HDBSCAN to cluster sequences. The authors also provide some realistic scenarios to prove the capability of their methods on visualization and interpretation of epitranscriptomic data. In general, I only have the following 5 questions or concerns that need authors to reply or polish their manuscripts through some extra analyses.

Q1: It's dangerous to over-interpret the coordinate of non-linear dimensional reduction UMAP or t-SNE. The ability of capture global data structure of both methods highly depend on the initialization (<https://www.nature.com/articles/s41587-020-00809-z#ref-CR2>). There are also some debates on if there exists extensive distortions on 2D embedding space (<https://www.biorxiv.org/content/10.1101/2021.08.25.457696v4.full.pdf>). Using distance defined in 2D space as the metric may not be sufficient and even could be wrong.

We thank the reviewer for this comment. We totally agree with the reviewer's opinion that initialization of UMAP and t-SNE may affect the decomposition output, and 2D space might not be the best solution for the decomposition.

To assess the potential influence of various initialization methods, we tested UMAP and t-SNE on different initialization methods with a simulated dataset. This dataset contains 12 randomly chosen transcription factor motifs from JASPER. The lengths of the motifs vary from 6 bp to 15 bp, and the numbers of motifs vary from 100 to 20,000. We also included 50,000 random noises. In total, this dataset has 109,850 different sequences. We found that, most motifs were correctly clustered on a 2-D plane by UMAP and t-SNE, no matter which initialization method is used (**Extended Data Fig. 1d**). The performances of UMAP and t-SNE depends on the uniqueness and occurrence of the underlying patterns. Longer and unique motifs, such as the ESR2 motif, can be well separated from other sequences, even when their occurrence is relatively low. Shorter motifs may also be separated from others when they occur more frequently.

In conclusion, within the context of our specific scenario, a 2-D space appears to be adequate for achieving satisfactory performance with UMAP and t-SNE, while the choice of initialization method exhibits limited influence on the decomposition process. We postulate that the minimal impact of dimensionality and initialization methods in our motif analyses may be attributed to the simplicity of our input data. However, in more intricate scenarios such as single-cell analysis, their effects may become more apparent. In addition, it is possible that UMAP and t-SNE can be improved by higher dimension space or a specific initialization method. We have incorporated these additional results and discussion into the revised manuscript (**Page 6, Lines 137-141; Page 18, Lines 490-492**).

“Additionally, we tested UMAP and t-SNE on different initialization methods with a simulation dataset containing 109,850 sequences (Supplementary Note 1). We found

that most motifs can be correctly clustered by UMAP and t-SNE, no matter which method is used (Extended Data Fig. 1d).”

“Moreover, although 2-D spaces are used in both decomposition and clustering in this study, for some complicated datasets, it is possible to project the k-mer patterns onto a 3 or higher dimension space for further clustering.”

Extended Data Fig. 1d. The performance of UMAP and t-SNE with different initialization methods. The simulation dataset containing 12 motifs. Two initialization methods in UMAP (spectral and random) and three initialization methods in Open-tSNE (spectral, random, and PCA) were tested. PCA initialization is unavailable for the current version of UMAP, although the parameter is shown in the manual. Different motifs were shown in colors. The numbers of motifs were shown in parentheses.

Q2: Following Q1, I think the key goal (correct me if I am not right) in the first step is subtyping. Giving a good representation on 2D space is not the ultimate goal. But finding latent subtypes of sequence (modification) is the goal. It is also the same thing in cell type identification of single cell analysis. Therefore in single cell gene expression analysis, the normal routine of clustering is to first use Top-K principal components (PCs) to build neighborhood graphs and use graph-based methods, e.g. leiden and louvain to cluster cells. A DBSCAN based clustering on t-SNE coordinate is used in a very old version of Seurat (V2 I think), but has been discarded. So I am wondering how the author performs louvain and leiden clustering (I didn't find this part in their github repo <https://github.com/SYSU-zhanglab/iMVP/tree/main/Benchmarking>). The author needs to show that using the UMAP+HDBSCAN method is indeed better than KNN graph+leiden or louvain clustering. In general, I hope the author gives a more rigorous benchmark to support that UMAP+DBSCAN is the best clustering algorithm. But downstream data visualization could be UMAP or t-SNE, I merely question about their clustering algorithm.

To benchmark the clustering performance, one could use other metrics specifically design for clustering analysis, e.g. Adjusted Rand Index (ARI). To compare your unsupervised learned label and ground truth label (Type I and Type II).

We are grateful for the reviewer for these comments and suggestions. We agree with the reviewers that motif clustering and demonstration can be separated. To address the question about the performance difference between UMAP+HDBSCAN and KNN graph+Leiden, we have included an analysis of ARI as suggested by the reviewer. We found that, in simple cases such as the fly dataset illustrated in **Fig. 1**, KNN

graph+Leiden had a better performance (**Extended Data Fig. 2b**). However, when the dataset contains random noise, such as in the case of the large simulation dataset, UMAP+HDBSCAN is a better choice (**Extended Data Fig. 2c-e**).

It is worth noting that HDBSCAN offers two clustering modes: "normal clustering," which leaves data points unclassified, and "soft-clustering," where all data points are assigned to a cluster based on maximum likelihood. Both HDBSCAN (with unclassified clusters) and HDBSCAN (soft-clustering) had better ARI than Leiden in this case. We have revised the manuscript to provide further clarification on this matter (**Page 6, Lines 149-156**).

Additionally, we would like to apologize for the delayed synchronization of the notebooks. In our analysis, we conducted Leiden and Louvain using Scanpy: the KNN distance matrix was obtained through the UMAP API, and the graph for Leiden and Louvain was constructed using igraph. Then the Python packages leidenalg and louvain were employed.

Extended Data Fig. 2b. The ARI scores in the fly embryo m5C dataset.

Extended Data Fig. 2c-e. The clustering results and ARI scores of HDBSCAN, HDBSCAN (soft-clustering), and Leiden on the large simulation dataset. For Leiden, resolution parameter was set to 1 in this analysis.

Q3: The motifs type divergence across species need to be clearer. From Fig.3g, it's hard to tell the motif preference divergence between fly and vertebrate Type I site. I suggested the author need to fully describe the motif preference in vertebrate type I site, to make comparison with fly. Or use some statistical metrics, e.g. KL divergence, to illustrate the difference in motif state distribution.

We thank the reviewer for this suggestion. As the reviewer suggested, we used MotifComparison function within MotifSuite to calculate the KL divergences between Fly Type I motif subtypes and Type I motifs found in vertebrates. The results of this analysis align with the observations made in the iMVP analysis (**Fig. 3i**). We found that vertebrate Type I motifs exhibited a higher degree of similarity to the CRGNR subtype in Fly, with KL divergences ranging from 0.071 to 0.11. In contrast,

for the other two subtypes, the KL divergences between fly and vertebrate motifs ranged from 0.11 to 0.24. We have included these results in the revised manuscript (Page 11, Lines 302-304).

Fig. 3i. Pairwise comparison of Type I motifs (+1 to +5 positions) in different species. Metrics: KL divergence, lower values mean two motifs are more similar.

Q4: The authors claim that clustering motifs to have global motif visualization of the inputs may facilitate the identification of false positives. In general, it is, however, in the main text, those false positives belong to the known sequence error or artifacts, but in some cases without prior knowledge, how do we know a new sequence feature is a biological meaningful “feature” but not artifacts?

We appreciate the valuable comment from the reviewer. We acknowledge and agree with the reviewer's point that iMVP, by itself, can only identify enriched clusters and does not determine whether these clusters are false positives or genuine features. The primary utility of iMVP lies in its ability to highlight enriched signals that may be overlooked during typical mapping and filtering processes. Thus, further experimental investigation is crucial to discern whether these identified clusters represent significant biological features or are merely artifacts. We have clarified this aspect in the revised manuscript. (Page 13, Lines 335-337).

“Notably, iMVP itself can only identified enriched clusters, and additional experiment and/or knowledge is required to determine whether these clusters are false positives or not.”

Q5: Following Q4, What’s more, in the clustering stage, some newly identified clusters may not be meaningful, and clustering algorithms (including HDBSCAN) will partition data even in cases where there is only uninteresting random variation present. Therefore, in clustering there also exists a risk of introducing new artifacts, so assessing how much the clustering results can change due to uninteresting random variation is very important. The author needs to provide some methods to examine the statistical significance of each cluster, or statistical possibility of under- or over-clustering (<https://www.biorxiv.org/content/10.1101/2022.08.01.502383v1.full.pdf>). That would make the results more reliable but not highly dependent on how much we know about biology.

We thank the reviewer for this suggestion. To effectively address the question, it can be divided into two sub-questions: (1) tuning the parameters to achieve appropriate clustering results, and (2) identifying meaningful signals after "proper" clustering, without the knowledge of ground-truth.

For sub-question (1), we have utilized the Density Based Cluster Validity (DBCv), which is implemented in the HDBSCAN API, to search for suitable parameter combinations. Our findings indicate that the clustering results are generally robust across different parameter settings (see https://github.com/SYSU-zhanglab/iMVP/tree/main/HDBSCAN_optimization_by_DBCv). Additionally, users have the flexibility to choose between normal HDBSCAN clustering or soft clustering approaches. This option allows for mitigating potential over-clustering challenges when working with noisy data. Regarding the Leiden algorithm, density-based metrics are not applicable. Therefore, we suggest using the Significance of Hierarchical Clustering (SHC) metric, as recommended by the reviewers, to assess the clustering quality for Leiden. These details have been incorporated into the revised method section of the manuscript (**Page 35, Lines 791-794**).

For sub-question (2), we have used the Simple Enrichment Analysis (SEA) tool within MEME toolkit to calculate p-value, q-value, and E-value for the motifs discovered by iMVP. These measures provide valuable information into the significance and enrichment of the motifs within the analyzed dataset. All measures have been presented in a new table (**Supplementary Table 8**).

Reviewer #3 (Remarks to the Author):

The authors describe a new approach to analyze motifs in base modification sequencing data using 2D projection and clustering of one-hot encoded 21-mers. The method is an interesting application of dimensionality reduction and clustering, and the authors demonstrate its potential utility for data exploration. The most convincing finding points to NSUN5 and Nop2 modification of mRNA sites. However, a number of other claims are overstated or require further validation.

Major comments

1. Direct RNA nanopore sequencing is error prone. What are current estimated rates for base calling errors and to what extent would this affect the data included in these analyses? The authors state, for example, “Other small clusters were centered by non-A bases (Extended Data Fig. 3b-e) and were less likely to be m6A signals; thus, they were not included in the analysis.” (p 7).

We are grateful to the reviewer for bringing attention to this point. Nanopore direct RNA sequencing has been reported to exhibit read accuracies of approximately 90%. In our study, we utilized the m6A sites called from a published paper by Pratanwanich, P.N. et al. (Nature biotechnology 39, 1394-1402, 2021). The authors of that paper estimated a false positive rate of approximately 10% for the identified m6A sites.

These false positives can result in two distinct patterns: (1) random noise, which corresponds to random sequences, and (2) artifacts, which may exhibit certain patterns but do not represent genuine signals. In the iMVP analysis, random noise tends to be sparsely distributed in the 2D space, leading to a lack of enrichment during clustering. On the other hand, artifacts might manifest as clusters, such as those centered around non-A bases. And these artifact clusters can potentially be removed based on our existing knowledge about m6A. Consequently, the impact of false positives on deducing the genuine m6A clusters may be limited.

2. m6Am primarily follows 7-methylguanosine caps at the 5' ends of mRNA (see Mauer et al., 2016; Boulias et al., 2019; etc.). When the authors claim that “The remaining 5% of sites were within the CAG motif, which is an m6Am-like motif” (p 7) -- 1) do they have a citation for m6Am in CAG motifs? 2) What fraction of this cluster of sites was at the 5' end of mRNA? Does further exploring the hypothesis that these sites are m6Am modified make sense given their location or could these sites be ruled out as true m6Am sites based on non-5' location alone?

We thank the reviewer for the comment.

For question 1), we have corrected the description from CAG to CAR in xPore analysis (**Fig. 2**) and CAB (**Fig. 6**) based on the software calculation. There are 3 descriptions about the m6Am motif: BCA (Boulias et al., 2019), BCA (Sun et al., 2019), and BBCABW (Koh et al., 2019). The motifs we identified were consistent with these references, we have included the citations in the revised manuscript (**Page 9, Line 218**).

For question 2, we have examined the locations of m6Am-like and m6A sites relative to transcription start site (TSS).

For Fig.2 sites, despite the CAR motif resembles previously reported m6Am motif (BCA (B=C/G/U) or BBCABW, as METTL3 is exclusively associated with m6A modification, these sites are likely to be false positives. Consistently, when examining the locations of CAR sites relative to TSS, no enrichment around TSS was observed (Extended Data Fig. 6f). For Fig. 6 sites, in line with previous observations, the m6Am-like sites identified in Cluster #5 had the strongest enrichment around TSS; in contrast, the m6A sites in Cluster #1 to Cluster #4 had significantly weaker enrichments (Extended Data Fig. 11c). we have included these in the revised manuscript (Page 9, Lines 218-222; Page 15, Lines 395-397).

Extended Data Fig. 6f. Histograms showing the distributions of the distance between m6A/m6Am sites and the nearest TSSs. Cluster #1 was m6A cluster and cluster #2 was CAR motif sites. m6Am sites identified by miCLIP data (Linder et al. Nat Methods 12, 767–772 (2015)) were used as positive control.

Extended Data Fig. 11c. Histograms showing the distributions of the distance between m6A/m6Am sites and the nearest TSSs. Cluster #1 to #4 were m6A clusters and cluster #5 was m6Am-like cluster. m6Am sites identified by miCLIP data were used as positive control.

3. “Taken together, iMVP assists in the identification of a previously unknown trans-regulatory evolution between vertebrate and fly NSUN2 proteins” (p 10)

This seems like unnecessary jargon to describe results suggesting that homologous enzymes from different phyla have slightly different motif preferences. What is “trans-regulatory evolution”? Have the authors verified that the differences they see are not due to differences in the prevalence of the motifs in each species’ transcriptome rather than differences in the specificities of the enzymes?

We thank the reviewer for the comment. “Trans-regulatory evolution” means that the Type I motif differences among different species are attributed to divergences in the NSUN2 protein. We agree with the reviewer that it is an unnecessary jargon and we have removed this term in the revised manuscript.

To validate that the observed differences were not due to variations in motif prevalence across species, we conducted experimental verification. Specifically, we cloned four different NSUN2 genes from human, mouse, zebrafish, and fly and expressed them in NSUN2 knockout HeLa cells. The G-enrichment pattern at +1 to +5 positions of the motifs and the pairwise KL divergence between the motifs found in this experiment is consistent the observation made across multiple species (Fig.3j). These findings provide confirmation of our claim (Pages 11-12, Lines 304-307).

Fig. 3j. Pairwise comparison of Type I motifs found in HeLa cells and those methylated by exogenously expressed NSUN2 from human, mouse, zebrafish, and fly in NSUN2 knockout HeLa cells.

4. “These observations highlight the ability of iMVP to pinpoint false positives that were unaddressed by traditional filtering steps.” (p 10)

What were filtering parameters and how many true sites would have been lost if filtering was adjusted to exclude these sites? Can the authors include a ROC plot or report area under the curve depending on filter threshold?

We thank the reviewer for this comment. No specific filtering parameters were employed in this analysis. According to the iMVP analysis, clusters #4-7 were identified as false positives, and the corresponding sites within these clusters were considered as such and subsequently removed. As a result, we are unable to generate a ROC plot accordingly. To assess the potential loss of true positive (TP) sites, we conducted an analysis using wild-type and NSUN6/NSUN2 double knockout HEK293T cells. In this cell line, a total of 356 sites were identified, with 328 sites belonging to clusters# 1-3, and 28 sites belonging to clusters #4-7. Notably, all 28 sites in clusters #4-7 were determined to be false positive sites, as they exhibited

methylation in both wild-type and knockout cells. Notably, cluster #4-7 exhibited significantly lower levels of methylation in wild-type cells compared to cluster #1-3 (Fig. R1), further supporting their classification as false positives. Consequently, the removal of clusters 4-7 did not result in any loss of TP sites in this case.

Fig. R1. Boxplot showing the methylation of Cluster#1-3) and Cluster#4-7 sites.

5. “Interestingly, we found that the two centroids of repetitive non-Alu sites were located in poly(U)-tract and poly(A)-tract, respectively (Fig. 7d), which were mainly from LINE and LTR repeats (Fig. 7f). For nonrepetitive sites, the centroid cluster was constituted by the canonical Position -1 G-depletion motif (Fig. 7e). Taken together, these analyses indicate that iMVP can handle extremely large datasets and expand our understanding of the sequence features of A-to-I RNA editing sites.” (p 15)
 How are the authors confident that these sites are true positives?

We thank the reviewer for the comment. To verify the authenticity of the non-Alu sites located within the poly(U)-tract and poly(A)-tract regions, we conducted an analysis using ADAR1 knockout HEK293T cells. HEK293 cells predominantly express ADAR1 and exhibit low levels of ADAR2 expression. We found that both sets of sites had decreased editing levels (Fig. 7g). We have included this result in the revision.

Fig. 7g. Boxplots showing the editing level changes of editing sites located in the Ploy(U)-tract and Ploy(A)-tract between wild-type and ADAR1 knockout HEK293 cells. We required that the sites were covered by at least 10 reads in both samples. P values were calculated by the paired samples Wilcoxon test. ***, $p < 0.001$.

6. “the cluster output may vary with different parameters, especially when extracting clusters from the condensed density tree.”

To what extent does the cluster output vary? Also, UMAPs are based on a stochastic algorithm. Between this and sensitivity to parameter selection, how reproducible are results?

We thank the reviewer for these questions. The degree of variation in the output depends on the complexity of the datasets. In simple datasets, clustering often exhibits minimal variation; however, in more complex datasets that contain highly similar motifs, selecting an appropriate cluster size threshold becomes challenging as it affects the splitting of decomposition results based on density. In cases where similar patterns occur closely on the 2D plot, a high cluster size threshold may lead to the neglect of splitting these closely located motifs. For example, in **Figure 3** of our study, the Type I and Type IV m5C motifs can be grouped together without considering the local density. To address this issue, we have implemented an updated hyperparameter optimization method (see https://github.com/SYSU-zhanglab/iMVP/tree/main/HDBSCAN_optimization_by_DBCV), which is discussed in the updated notebook accompanying our study.

To assess the robustness of UMAP, we conducted several tests on various aspects. These included evaluating initialization methods (**Extended Data Fig. 1d**), random seeds (**Extended Data Fig. 1e**) and metrics (**Extended Data Fig. 1f**). The results showed that UMAP exhibits robustness across different parameter settings in general. We have included these results in the revised manuscript (**Page 6, Lines 137-141**).

Extended Data Fig. 1d. The performance of UMAP and t-SNE with different initialization methods. The simulation dataset consists of 12 motifs. Two

initialization methods in UMAP (spectral and random) and three initialization methods in Open-tSNE (spectral, random, and PCA) were tested. PCA initialization is unavailable for the current version of UMAP, although the parameter is shown in the manual. Different motifs were shown in colors. The numbers of motifs were shown in parentheses.

Extended Data Fig. 1e-f. Testing UMAP with different random seeds (e) or metric functions (f).

7. Why did the authors choose to encode 21-mers rather than another length of motif and does this choice affect iMVP performance?

We thank the reviewer for the comment. In this study, we focus on the analysis of RNA modification motifs, which typically consist of relatively shorter sequences. For instance, the NSUN5 motif spans 11 nucleotides, while the METTL3/METTL14-dependent m6A motif is approximately 5 nucleotides long. Given this, utilizing 21-mers is both sufficient and computationally efficient for our analysis.

In principle, iMVP should be able to handle inputs of diverse lengths. To test this, we examined the m5C datasets in our study. We found that when employing k-mers spanning from 17 to 51 nucleotides, distinct m5C motifs could be effectively distinguished (**Extended Data Fig. 3**). We have included this in the revised manuscript (**Page 7, Lines 162-166**).

Extended Data Fig. 3. The impact of k-mer length selection on iMVP.

(a-b) iMVP was performed on 5, 8, 10, 15, 20, 25, and 50 nt flanking sequences (11 to 101 mers) of the m5C sites found in fly embryos (a) or Noc-treated HeLa cells (b).

Minor comments

1. In the introduction, the authors state, “writers of m6A and A-to-I editing are misregulated in multiple types of cancers and have been identified as promising therapeutic targets of cancers, with related cancer treatments under development”. Can they expand or offer citations on which cancer treatments are actually under development targeting m6A or A-to-I editing writers and their stage of development?

Briefly consulting a 2021 review from Yang et al., (“RNA methylation and cancer treatment” <https://doi.org/10.1016/j.phrs.2021.105937>) does not identify any therapies targeting writers approaching clinical trials.

We appreciate the reviewer for bringing this to our attention. We have expanded our description and offer citations to point out the stages of development (**Page 3, Lines 50-55**).

“For example, the writers of m6A and A-to-I editing are misregulated in multiple types of cancers and have been identified as promising therapeutic targets of cancers³⁻⁵, with related cancer treatments in pre-clinic or pre- investigational new drug (IND) stages. Catalytic inhibitor of METTL3 STM2457 leads to impaired engraftment and prolonged survival in various mouse models of AML5. ADAR1p150 inhibitor Rebecsinib prevents malignant A-to-I editing-mediated leukemia stem cell self-renewal in completed pre-IND studies⁶.”

2. “the similarities of the sequences were therefore represented by the Euclidean distances in the projections” implies that these distances are accurate representations of similarity. However, see for instance Chari and Pachter (bioRxiv, 2022) and their references, showing that distances are inevitably distorted with dimensionality reduction of many points. I would suggest changing “represented” to “approximated”.

We thank the reviewer for this suggestion. We have changed “represented” to “approximated” as the reviewer suggested.

3. “also true for more complicated m5C data from nocodazole (Noc) treated HeLa cells” (p. 5) For readability, please introduce here the expected or measured effects of nocodazole on m5C. How is this data more complicated?

We thank the reviewer for pointing this out. With nocodazole (Noc) treatment, at least four m5C writers were activated to methylate mRNAs, thus at least four motifs, including two minor motifs were present in this dataset. We have incorporated an additional sentence in the manuscript to introduce the effects of nocodazole on m5C (**Page 6, Lines 133**).

“These findings were also true for more complicated m5C data from nocodazole (Noc) treated HeLa cells, in which at least 4 motifs, including two minor motifs, are present (**Extended Data Fig. 1b**).”

4. “Globally, we found that only a small set of m6A and m6Am sites were overlapped among methods (Extend Data Fig. 7a), although the same cell type was used in all studies. This observation suggests that each method only captured a subset of the methylated sites.”

Alternatively, each method captures many false positives or m6A levels vary in this cell type. Do FP rates differ among these methods? Which cell type was used to generate these datasets?

All the studies utilized HEK293 or HEK293T cells. Additionally, m6A-label-seq and MAZTER-seq included sites in HeLa or hESC, respectively. Due to the absence of ground-truth site lists, these methods were unable to accurately estimate the rates of

false positives. We agree with the reviewers that the potential variability in false positive rates may contribute to the low overlap observed between methods. Furthermore, recent studies employing more sensitive single-base resolution techniques like GLORI (Liu et al., *Nat Biotechnol* 2023) have indicated the presence of over 170,000 sites in a single cell type. This suggests that previous methods may have captured only a subset of the methylated sites, particularly those with lower levels of methylation. We have included a sentence in the manuscript to highlight this point (**Page 14, Lines 385-387**).

“Furthermore, the potential variability in false positive rates among different methods may also contribute to the observed low overlap.”

5. “Compared with MAZTER-seq, signals from m6A-REF-seq had much lower m6A-peak enrichment (Fig. 6e), suggesting that the addition of FTO treatment might introduce noise.”

A table in the main text describing the differences among these methods would be helpful.

We thank the reviewer for this suggestion. We have now revised Fig. 6a to effectively depict and explain the differences among methods utilized in our study.

REVIEWERS' COMMENTS

Reviewer #1 (Remarks to the Author):

After thoroughly perusing the revised manuscript and its supplementary materials, I acknowledge that the authors have effectively addressed most of the initial concerns raised by the reviewers. There are, however, a few minor issues that need resolution to ensure the manuscript's reliability, after which it would likely be suitable for publication:

- Figures 3i and 3j should retain the upper triangles, given the asymmetry of the Kullback-Leibler (KL) divergence.
- It is crucial to include a negative control in Figure 11c. Data handling or the sequencing method employed on a dominant dataset might inadvertently introduce a bias toward the 5' end.

Reviewer #2 (Remarks to the Author):

In general, the authors have addressed most of my questions. But I have one minor comments regard on Q5.

In Q5, as authors response, they have provided several options to improve motif clustering results. I would like to see some examples that these metrics actually could realise "mitigating potential over-clustering challenges when working with noisy data". My expectation is that, after quality control step, it would actually filter the clusters come from data noise. Please including those results as part of the benchmark study of your methods

Some statistical methods that already been proposed to overcome this issue, I suggest authors could check "<https://www.nature.com/articles/s41592-023-01933-9#Sec9>"

Reviewer #3 (Remarks to the Author):

Thanks to the authors for their response and for addressing most of my concerns. The manuscript has improved, and the additional experiments with NSUN2 complement the story nicely.

I have the following minor comments to further clarify the manuscript before publication.

- 1) It is unclear why the authors refer to other methods for motif detection with published algorithms as black boxes in the introduction and discussion, or what they mean by global view in the statement, "Such a process is a black box, and we have no grasp of the global view of the dataset or the levels of noise within the dataset." Most motif finders provide, for instance, frequencies of motif detection across datasets, which could constitute a "global view".
- 2) "With this knowledge, we may, in principle, assign the writers of each authentic modification site even without experimental validation." I believe that the authors are trying to convey that if the motif of

a site matches the known target motif for a writer, then one can hypothesize that that writer targets that motif. However, it would help to specifically note that prior knowledge of writer target motifs is required.

3) “in consideration of the performance (Fig. 1c) and robustness (Extended Data Fig. 1e-f) of the algorithms, we selected UMAP as the best algorithm for dimension reduction.” What is the metric for robustness? The Fig 1b and Extended Data Figure 1a/b/d suggest similar performance between UMAP and t-SNE, and the cited Extended data Fig. 1e-f show only UMAP. I'm not suggesting that the authors redo their work using t-SNE, but they should revise their statement here in light of the fact that performance seems to be comparable with t-SNE.

4) It would clarify the text to better explain what a winscore does at first mention (e.g. based on the reported formula, it quantifies the enrichment of modified reads within a window relative to gene expression and modified reads across the entire gene). It is unclear from the methods how the winscore is calculated for nanopore data if there is no IP.

5) The subfigures in Extended Data Figure 5 are unlabelled, and the metric reported in (f) corresponds to the formula for precision rather than accuracy.

6) Please show the TSS distances in Extended Data Figure 7, as elsewhere.

7) “Since the majority of reported RNA modification motifs are within a range of 15 nucleotides (nt)” requires citation.

8) “canonical CAB motif for m6Am sites” (line 394) -- the canonical motif is correctly reported earlier as BCA or BBCABW. CAB is similar, but not canonical.

Reviewer #1 (Remarks to the Author):

After thoroughly perusing the revised manuscript and its supplementary materials, I acknowledge that the authors have effectively addressed most of the initial concerns raised by the reviewers. There are, however, a few minor issues that need resolution to ensure the manuscript's reliability, after which it would likely be suitable for publication:

- Figures 3i and 3j should retain the upper triangles, given the asymmetry of the Kullback-Leibler (KL) divergence.

We thank the reviewer for this comment. We apologize for the unclear depiction of the KL divergence in our manuscript. The term “KL divergence” in the manuscript refers to the “Average KL divergence”, which quantifies the divergence between query and reference comparisons in both directions.

In motif analysis, it is common to use “Average KL divergence” rather than two KL divergences to represent the motif similarity. For example, R package TFBStools (https://bioconductor.org/packages/devel/bioc/vignettes/TFBSTools/inst/doc/TFBSTools.html#ref-linhart_transcription_2008) and MotifComparison (<http://bioinformatics.intec.ugent.be/MotifSuite/usemotifcomparison.php>).

To prevent any ambiguity, we have now used the term “Average KL divergence” in our manuscript.

- It is crucial to include a negative control in Figure 11c. Data handling or the sequencing method employed on a dominant dataset might inadvertently introduce a bias toward the 5' end.

We thank the reviewer for this suggestion. Since m6A sites are known to be not enriched around 5' ends, they can serve as a negative control. Compared with cluster #5, which has >0.5 site density around TSS, the site densities around TSS of clusters #1 to #4 (m6A sites) are <0.05. We have revised the figure legends to point it out.

Reviewer #2 (Remarks to the Author):

In general, the authors have addressed most of my questions. But I have one minor comments regard on Q5.

In Q5, as authors response, they have provided several options to improve motif clustering results. I would like to see some examples that these metrics actually could realise "mitigating potential over-clustering challenges when working with noisy data". My expectation is that, after quality control step, it would actually filter the clusters come from data noise. Please including those results as part of the benchmark study of your methods

Some statistical methods that already been proposed to overcome this issue, I suggest authors could check "<https://www.nature.com/articles/s41592-023-01933-9#Sec9>"

We thank the reviewer for this suggestion. We agree with the reviewer that in certain instances, such as when employing Leiden, the Significance of Hierarchical Clustering (SHC) can be used to assess clustering quality. We have now included the reference to emphasize this.

Meanwhile, we've explored the SHC method in detail and found it may not align with our data in other cases, due to its reliance on a specific distribution (e.g., Gaussian) for generating data representing the null hypothesis. Given our context with DNA/RNA sequences, determining an appropriate distribution is challenging. This lack of a statistical model prevents us from generating simulated data to assess enrichment significance. Our attempts to adapt SHC using the sc-SHC package in R to evaluate cluster significance were hindered by the fundamental disparity between our data structure and single-cell sequencing data.

Regarding over-clustering issues, we face two main challenges: over-partitioning clusters and random noise-induced fake clusters. To address over-partitioning, we can estimate their similarity using parameters like KL divergence. As for random noise, fake motifs typically exhibit low base enrichment (entropy) and can be filtered out accordingly.

Reviewer #3 (Remarks to the Author):

Thanks to the authors for their response and for addressing most of my concerns. The manuscript has improved, and the additional experiments with NSUN2 complement the story nicely.

I have the following minor comments to further clarify the manuscript before publication.

1) It is unclear why the authors refer to other methods for motif detection with published algorithms as black boxes in the introduction and discussion, or what they mean by global view in the statement, "Such a process is a black box, and we have no grasp of the global view of the dataset or the levels of noise within the dataset." Most motif finders provide, for instance, frequencies of motif detection across datasets, which could constitute a "global view".

We agree with the reviewer that the results provided by other motif finders may constitute a global view as well. We have revised the introduction section and this claim is no longer present in the manuscript.

2) "With this knowledge, we may, in principle, assign the writers of each authentic modification site even without experimental validation." I believe that the authors are trying to convey that if the motif of a site matches the known target motif for a writer, then one can hypothesize that that writer targets that motif. However, it would help to specifically note that prior knowledge of writer target motifs is required.

We thank the reviewer for this suggestion. We have revised the introduction section as the reviewer suggested.

3) “in consideration of the performance (Fig. 1c) and robustness (Extended Data Fig. 1e-f) of the algorithms, we selected UMAP as the best algorithm for dimension reduction.” What is the metric for robustness? The Fig 1b and Extended Data Figure 1a/b/d suggest similar performance between UMAP and t-SNE, and the cited Extended data Fig. 1e-f show only UMAP. I'm not suggesting that the authors redo their work using t-SNE, but they should revise their statement here in light of the fact that performance seems to be comparable with t-SNE.

We thank the reviewer for pointing it out. We have revised the manuscript and this statement is no longer in the text.

4) It would clarify the text to better explain what a winscore does at first mention (e.g. based on the reported formula, it quantifies the enrichment of modified reads within a window relative to gene expression and modified reads across the entire gene). It is unclear from the methods how the winscore is calculated for nanopore data if there is no IP.

We thank the reviewer for this suggestion. We have included the definition of "winscore" upon its initial mention - “Winscore quantifies the enrichment of modified reads within a 50-nt window relative to gene expression and modified reads across the entire gene.”.

Notably, we used winscore methods based on illumine-sequencing to validate the nanopore data. Consequently, no winscores were computed based on the nanopore data.

5) The subfigures in Extended Data Figure 5 are unlabelled, and the metric reported in (f) corresponds to the formula for precision rather than accuracy.

Thanks for pointing this out. We have added the panel labels and corrected the (f) y-axis label.

6) Please show the TSS distances in Extended Data Figure 7, as elsewhere.

We have included the TSS distance plot in the revision.

7) “Since the majority of reported RNA modification motifs are within a range of 15 nucleotides (nt)” requires citation.

Thanks. We have added a citation accordingly: Wiener, D. & Schwartz, S. The epitranscriptome beyond m6A. Nat Rev Genet 22, 119-131 (2021).

8) “canonical CAB motif for m6Am sites” (line 394) -- the canonical motif is correctly reported earlier as BCA or BBCABW. CAB is similar, but not canonical.

Thanks, we have revised the sentence accordingly - “Cluster #5 was a CAB motif, similar to previously reported m6Am motifs.”